# BENCHMARKING LLM TOOL-USE IN THE WILD

**Peijie Yu**[1], **Wei Liu**[2], **Yifan Yang**[1], **Jinjian Li**[1], **Zelong Zhang**[1]
**Xiao Feng**[1], **Feng Zhang**[1]

[1]Tencent HY, [2]King's College London
{peijieyu@tencent.com}

## ABSTRACT

Fulfilling user needs through Large Language Model multi-turn, multi-step tool-use is rarely a straightforward process. Real user interactions are inherently **wild**, being intricate, messy, and flexible. We identify three key challenges from user behaviour: *compositional tasks* that demand efficient orchestration of tool-call topologies, *implicit intent* spread across dialogue turns that require contextual inference, and *instruction transition*, which mixes task queries, clarifications, and casual conversation, forcing LLMs to adjust their policies on the fly. Existing benchmarks overlook these behaviors, making the apparent progress of LLMs on tool-use spurious. To address this, we introduce **WildToolBench**, an LLM tool-use benchmark grounded in real-world user behavior patterns. Comprehensive evaluations of 57 LLMs reveal that no model achieves an accuracy of more than 15%, indicating a substantial gap in the robustness of LLMs' agentic ability. Controlled experiments and in-depth analyses further indicate that the real challenge for LLM tool-use lies not in artificially complex tasks, but in the wild nature of user behavior, emphasizing the need to reconsider the interactions among *LLMs*, *users*, and *tools*. The benchmark is publicly available at https://github.com/yupeijei1997/WildToolBench.

## 1 INTRODUCTION

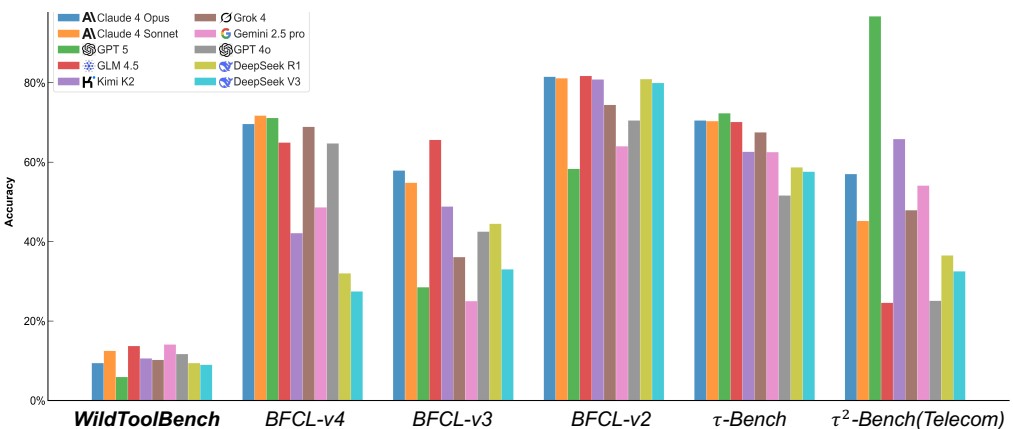

Figure 1: Session Accuracy comparison among tool-use benchmarks. See details in Appendix B.

Large language models (LLMs) are evolving rapidly, and agents built on them have become a promising direction (Google, 2024; DeepSeek-AI et al., 2025; Zeng et al., 2025; Yao et al., 2023; Liu et al., 2024a; 2026; Qian et al., 2024b; Qian et al.; 2024a). These agents interact with the real world through various tools, opening up new avenues for AI applications. Developing benchmarks that can evaluate the tool-use capabilities of large language models in a reliable way has become increasingly important.

Current mainstream LLM tool-use benchmarks follow a multi-turn, multi-step paradigm: LLMs function as assistants and engage in multi-turn dialogues with users to complete coherent tasks. Each

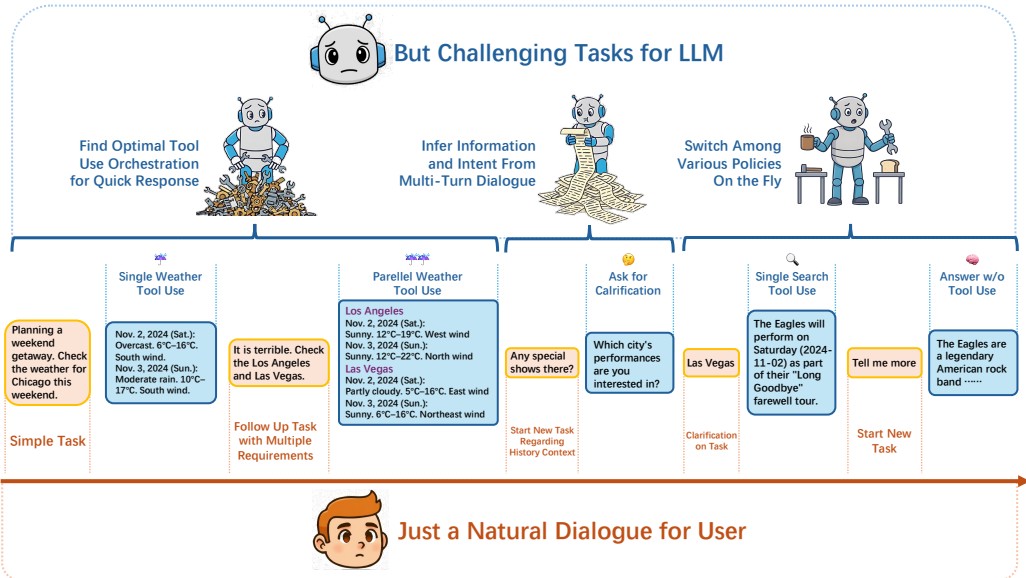

Figure 2: *WildToolBench* poses three characteristics that seem easy and natural for the user, but challenging for the LLM tool-use.

task typically requires multi-step tool-use. However, existing benchmarks (Huang et al., 2024a; Qin et al., 2024; Du et al., 2024; Yao et al., 2024; Ji et al., 2024b) are overly idealized and neglect the complexity of multi-turn, multi-step settings in real-world scenarios. From large-scale analysis of real user logs, we identify three salient properties of how human users employ LLMs to solve tasks with tools: 1) users tend to deliver **Compositional Tasks** that contain multiple simple requirements, demanding tool orchestration beyond simple chaining to respond on time. 2) Users' **implicit intention** is spread within dialogue, requiring LLMs to infer it from context. 3) In a conversation, users naturally **transition between different types of instructions**, such as task-giving, follow-up, explanation, and casual chatting modes, demanding LLMs to adapt their policies on the fly.

These three characteristics embody the design philosophy of *WildToolBench*, *"What truly challenges LLMs' tool-use capabilities is not artificially constructed complex scenarios, but simple yet realistic user behaviors"*, namely, the compositionality, vagueness, and variability of user instructions. In *WildToolBench*, through a carefully constructed data pipeline combined with human verification and annotation, we curate 256 scenarios with 1024 tasks. As shown in Figure 1, while prior tool-use benchmarks tend to be saturated, *WildToolBench* remains highly challenging. Our results show that even the most advanced language models struggle to achieve satisfactory performance, with most models reaching no more than 15% session accuracy. A further breakdown of experiments on 57 LLMs reveals that *in-the-wild* task settings severely degrade model performance, underscoring that the future evaluation of LLMs' agentic ability cannot rely on simple, idealized benchmarks but must instead account for the inherent complexity of real-world user behaviours.

## 2 RELATED WORK

LLM agents have emerged as a prominent research direction, with their core competency rooted in the ability to utilize external tools. Tool-use benchmarks have, to some extent, shaped the evolution of LLMs' agentic ability, from simple QA to multi-turn, multi-step, long-horizon autonomous tool-use. T-EVAL, UltraTool, and MetaTool(Chen et al., 2024; Huang et al., 2024a;b) assess various sub-capabilities of tool-use, but treat tool invocation as a simple question-answering task, which fails to capture the multi-turn interactive nature of the LLM agent loop. WorfBench and TaskBench(Qiao et al., 2025; Shen et al., 2024) took a step forward by introducing single-turn multi-step tool invocation and emphasizing planning capabilities, but are constrained by annotating only a single optimal path and relying on similarity-based metrics, which can be imprecise in evaluation. ToolBench, AnyToolBench, and StableToolBench(Qin et al., 2024; Du et al., 2024; Guo et al., 2024) also focus on single-turn

multi-step tool-use, but their proposed tasks are synthesized by LLMs and generally exhibit a low level of difficulty. On the other hand, BFCL-V1 and BFCL-V2 (Ji et al., 2024a) pioneered the evaluation of parallel tool-use but were still limited to single-turn scenarios. BFCL-V3(Ji et al., 2024b) introduced multi-turn evaluation and assessed the sequential multi-step capabilities of LLMs. However, its tasks lack semantic correlation, with each task being independent and identically distributed, and provided with complete intention and information, which is unnatural compared with real-world user behaviours. Therefore, $\tau$-Bench and $\tau^2$-Bench(Yao et al., 2024; Barres et al., 2025) introduce the design of LLM-as-User. To some extent, user simulators better approximate real environments (e.g., requiring an LLM agent to proactively ask questions rather than merely execute tool calls reactively). However, LLM-based simulation still diverges significantly from real user behavior. For instance, LLMs tend to behave in an unrealistically flawless manner, making tasks too easy to solve. Moreover, reliance on LLM simulation also leads to unstable evaluation results. Through a human-in-the-loop annotation process, *WildToolBench* explicitly incorporates three real user behaviors (compositional tasks, implicit intent, and instruction transition), thereby setting a new standard for evaluating LLM tool-use. A comparison between *WildToolBench* and previous benchmarks is provided in Table 1.

## 3 WILDTOOLBENCH

### 3.1 FORMULATION

We formalize the interaction between a user and an LLM as a **multi-turn dialogue**, denoted as

$$D = \{u_1, a_1, u_2, a_2, \ldots, u_N, a_N\}$$

where $u_i$ is the $i$-th user message and $a_i$ is the corresponding LLM assistant response. Within this $N$-turn dialogue, there are $M$ user tasks $\{g_1, \ldots, g_M\}$ which are scattered throughout the dialogue. For each user message $u_i$, there may exist a task $g_j$, and the LLM needs to detect the user's intention and solve the task in the response $a_i$. If solving this task requires tool usage, the LLM will first engage in a process of **multi-step tool invocation**, which can be regarded as the LLM conducting several rounds of interaction with the external environment (e.g., a local database or a MCP server), denoted as $T^j = \{a_1^T, e_1, a_2^T, e_2, ..., a_S^T, e_S\}$, where $a^T$ is the LLM's tool call, and $e$ is the corresponding environment feedback after executing this call. Once this $S$-step tool invocation $T^j$ is completed, the LLM gathers information from feedback and generates the user's response to the task with $a_i$.

In a real scenario, user intentions are varied in one dialogue session, and user messages are mixed with various types of tasks $g$, such as asking questions, requesting follow-ups, seeking improvements, explaining themselves, or just chatting. The LLM needs to apply different policies for correct reactions, which may include 1) LLM just replies without any tool usage ($S = 0$), such as in response to a task that needs clarification $g_{clarify}$ or a task that does not require a tool $g_{chat}$, 2) LLM adapts a single-tool invocation policy ($S = 1$) for a simple task $g_{single}$, or 3) LLM performs multi-step tool invocations ($S > 1$) for a hard task $g_{multi}$. From the LLM's perspective, the dialogue unfolds as a Markov Decision Process (MDP), where the state at each step is the full dialogue history (including $u$, $a$, $a^T$, and $e$), and the actions are the tokens that formulate different policies. Under this formalization, *WildToolBench* faithfully reflects the complexities and challenges inherent in applications for real-world users, where 1) the user task $g$ is compositional, consisting of multiple sub-requirements, necessitating effective tool orchestration. This implies that $T$ may be a tree rather than a simple chain-like execution. 2) User tasks $\{g_1, \ldots, g_M\}$ are contextually interrelated, requiring the LLM to uncover latent context from historical observations, including user messages $\{u\}$ and assistant messages $\{a\}$. 3) User intentions transition in each message $u_i$, and the LLM must switch its policies accordingly to give a correct response $a$.

### 3.2 DATA CURATION

The data curation pipeline of *WildToolBench* follows three steps as shown in Figure 3. First, we summarized seed application scenarios and user behavior patterns from large-scale real user logs, and iteratively rewrote and expanded them into 256 scenarios. Then, following ToolAlpaca (Tang et al., 2023), we collected more than 1,600 publicly available APIs from the internet, carefully verified and cleaned them into a tool set. For each scenario, we picked a reasonable tool subset and generated 4 tasks based on it. Finally, we employed GPT-4o (OpenAI, 2024a) to construct a multi-agent system simulating the roles of user and assistant, generating initial trajectories under the given task

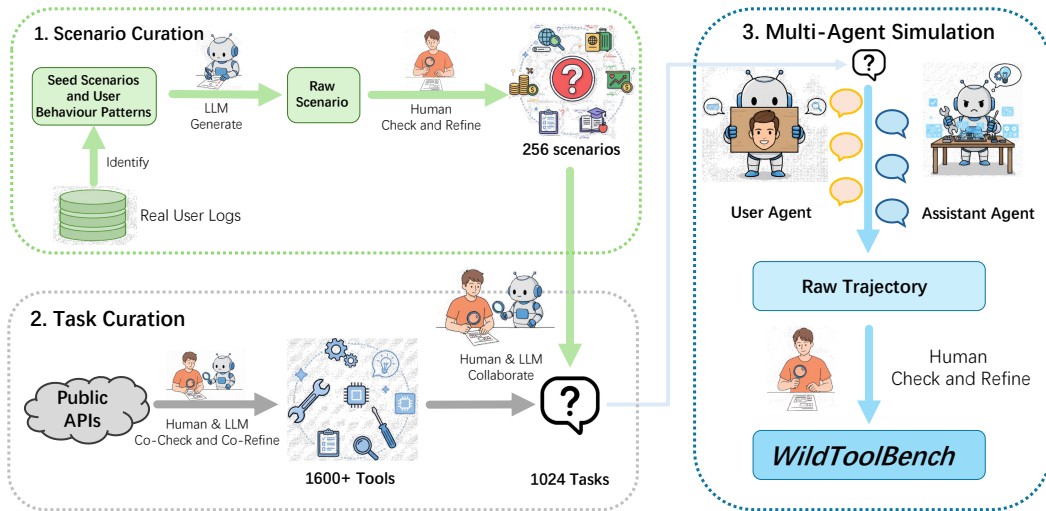

Figure 3: The data curation pipeline of WildToolBench.

and tool subset. Each tool invocation in the trajectory was manually examined and annotated as ground truth, producing the final dataset. The detailed process is described in Appendix §C. Each stage of the data curation pipeline involved manual annotation and validation to ensure accuracy and diversity. Furthermore, in the manual inspection of tasks, we emphasized three aspects: task compositionality (§3.3), contextualized intention (§3.4), and instruction transition (§3.5), reflecting the inherent complexity of real user behaviors. Finally, we present comprehensive statistics of *WildToolBench* in §3.6. The unique design of *WildToolBench* is highlighted in Table 1.

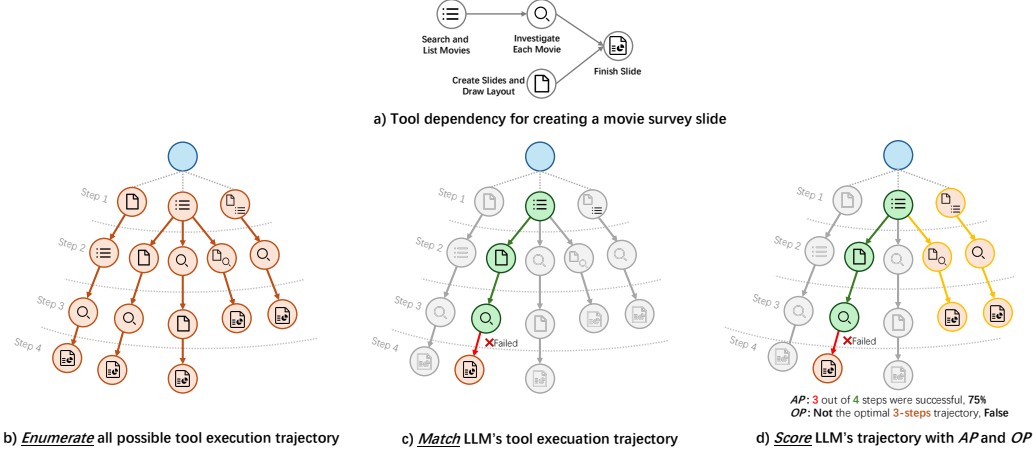

Figure 4: Visualization of the enumerate-match-score pipeline for evaluating the LLMs' tool orchestration ability in *WildToolBench*.

### 3.3 CHALLENGE 1: TOOL ORCHESTRATION FOR COMPOSITIONAL TASK

Real-world user instructions do not always present very hard tasks, but multiple simple requirements are combined into a single instruction. We meticulously constructed tasks under common scenarios (e.g., document operations or weather inquiries), but with compositional forms that better reflect real user instructions (e.g., searching for popular movies to generate a survey slide, or multi-city weather inquiries intertwined with travel planning). Compared with simple and well-defined tasks found in previous benchmarks, these are "in-the-wild" tasks that require an LLM to possess strong planning capabilities to identify tool dependencies and construct an efficient tool-calling topological graph, thereby improving TTFT (Time to First Token). To accurately measure whether an LLM can

effectively construct an efficient tool-calling topology, *WildToolBench* measures not only the final task accuracy but also more fine-grained metrics such as the optimal path rate and task accomplishment progress, in a simple three-stage manner: enumerate, match, and score.

**Enumerate** First, the adjacent tool dependencies are manually labeled by human experts. Then, we apply a depth-first topological sorting algorithm (see Appendix D for details) to generate all possible legal tool execution paths that obey the adjacent dependencies. Our approach enumerates all possible tool execution paths, rather than restricting to limited suboptimal paths (Qiao et al., 2025; Shen et al., 2024). Such an enumeration generates a decision tree set that considers all branching and parallel scenarios. For example, in Figure 4 a), the search-then-investigate branch and the slide branch can be executed in parallel, leading to five possible paths as shown in Figure 4 b).

**Match** Every time the LLM executes a tool, we use an incremental path matching strategy to locate this tool call in the previously enumerated decision tree set. Each tool call either terminates the path if mismatched or takes a step into the corresponding sub-tree.

**Score** By matching and locating the LLM's tool call in the enumerated decision tree set, we can evaluate the quality of the LLM's current tool execution topology. Whenever the tool executed by the LLM terminates or completes a path, we calculate whether this path has the minimum depth among all enumerated decision trees. If so, it indicates that this decision tree is not only valid but also possesses optimal efficiency, and we can calculate the *Optimal Path Rate* ($OP$ Rate). Furthermore, the LLM often fails on many tasks, with tool-calling nodes generated midway that do not fall within the valid decision tree set. We calculate the *Accomplish Progress Rate* ($AP$ Rate) based on the proportion of its successful nodes. These two fine-grained metrics, $OP$ and $AP$, are used to measure tool orchestration.

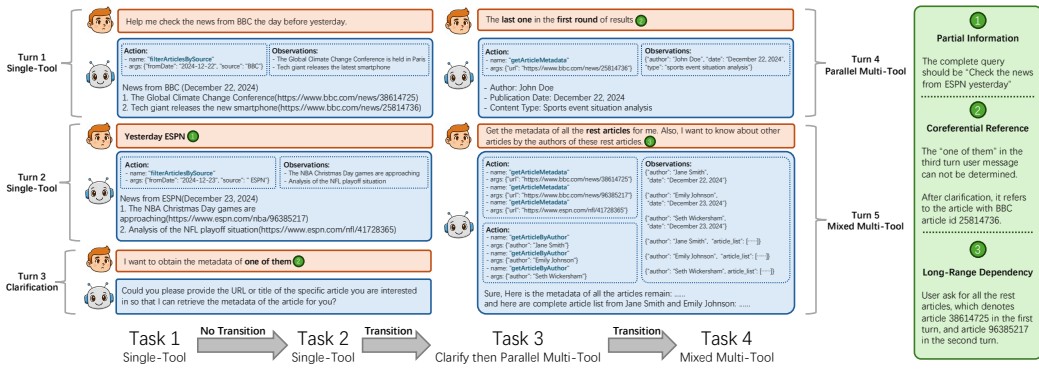

Figure 5: Examples for Challenges on Hidden Intention §3.4 and Instruction Transition §3.5. These challenges arise from the very nature of real user behavior: from the user's perspective, the interaction is **a coherent dialogue rather than a series of isolated task submissions**.

### 3.4 CHALLENGE 2: INFER HIDDEN INTENTION THROUGH DIALOGUE

Previous research (Chiang et al., 2023; Su et al., 2019) reveals that in sequential tasks, 80% of users follow up with additional questions and may modify or omit contextual information, which aligns with our observations. The LLM must infer the user's latent intentions from the multi-turn conversation, gather the necessary information and even proactively request clarifications. In *WildToolBench*, we utilize three strategies to construct tasks that demand multi-turn context inference, as shown in Figure 5: 1) **Partial Information**: The current user message $u_i$ contains only a subset of the information required to complete the task, while the omitted information is present in previous user or assistant messages $\{u_1, a_1, \ldots, u_{i-1}, a_{i-1}\}$. 2) **Coreferential Reference**: The current user message contains the full information, but the subject is expressed only via pronouns or ellipsis, referring back to entities mentioned in earlier user or assistant messages. 3) **Long-Range Dependency**: Similar to partial information, except that the missing information is located in distant dependencies; that is, $u_i$ depends on $\{u_1, a_1, \ldots, u_j, a_j\}$ with $i - j > 2$.

## 3.5 CHALLENGE 3: ADAPTABLE POLICY SWITCH FOR INSTRUCTION TRANSITION

When interacting with an LLM assistant, most users treat the interaction as a natural conversation rather than a series of independent task submissions. Users frequently initiate tasks across multiple turns, ask follow-up questions, provide explanations, engage in casual dialogue, and interrupt or resume tasks at will, continuously transition among different instruction types. As illustrated in Figure 2 and Figure 5, what appears to users as an ordinary conversation in fact involves multiple transitions. Such flexible and frequent instruction transitions require the LLM to adapt its policy appropriately, making suitable choices among strategies such as tool-use, direct question answering, or proactive inquiry.

In constructing *WildToolBench*, we categorize all tasks into four types: tasks solvable with a single tool call ($g_{single}$), tasks requiring multiple tools and multi-step calls ($g_{multi}$), conversational or tool-free queries ($g_{chat}$), and tasks that require the assistant to ask for clarification ($g_{clarify}$). For each scenario, we carefully curated the proportions of these four task types as well as their switching frequency, ensuring that *WildToolBench* faithfully reflects the phenomenon of instruction transition observed in real user behavior. This setup benchmarks LLMs' ability to accurately track evolving user intentions in natural dialogue and generate appropriate responses.

## 3.6 STATISTICS

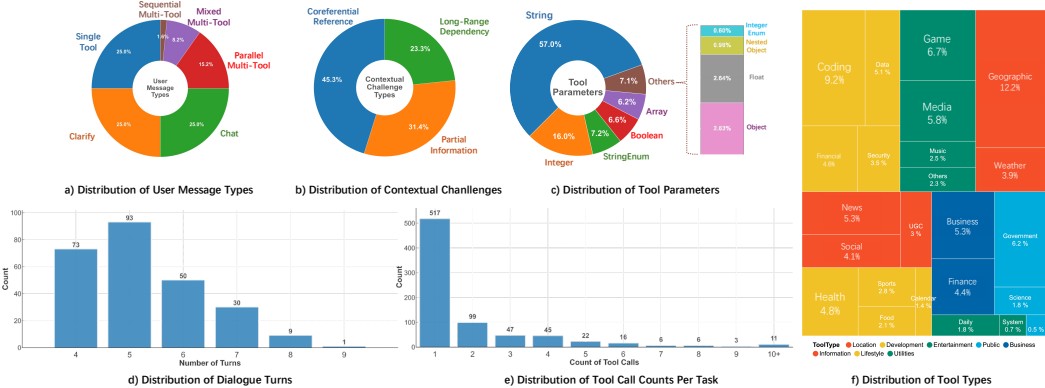

Figure 6: Key statistics for WildToolBench.

Figure 6 presents detailed statistics of *WildToolBench*. We constructed 256 scenarios, each consisting of a multi-turn dialogue with four user tasks, resulting in a total of 1,024 tasks. We evaluate whether the LLM generates the ground-truth tool calls within the dialogue, measuring both task-level accuracy and session accuracy, i.e., whether all four tasks in a dialogue are correctly completed.

The key observations from these statistics are as follows: 1) the four task types ($g_{single}$, $g_{multi}$, $g_{chat}$, $g_{clarify}$) and various forms of hidden user intention are well balanced, ensuring diversity and challenge within each dialogue; 2) tool parameter types are highly diverse, and the tool type covers 8 major categories and 24 subcategories, all of which correspond to commonly encountered real scenarios; 3) the average dialogue length is 5.27 turns and the average number of tool-call steps is 1.92, which is significantly higher than BFCL (3.75 turns, 1.68 steps). We highlight the wild nature of *WildToolBench* in Table 1.

## 4 EXPERIMENTS

We present a detailed experiment by benchmarking **57** mainstream LLMs on *WildToolBench*, ranging from proprietary to open-source LLMs, from general to specialized models, and from instruction-tuned to large reasoning models. The experiments and analysis are organized as follows: §4.1 gives an overview of benchmarking results and key takeaways, §4.2 investigates how well LLMs can orchestrate tool calls to handle the compositional user instructions in *WildToolBench*, §4.3 examines the inference ability of LLMs when users omit or hide their intentions and information across multiple turns in

Table 1: Comparative analysis of the *WildToolBench* against other tool-use benchmarks.

| Benchmark | Contextual Multi-Task | Hidden Info in Context% | User Instruction Transition% | Sequential Tool-Use | Parallel Tool-Use | Mixed Tool-Use |
|---|---|---|---|---|---|---|
| **WildToolBench** | ✓ | **100%** | **100%** | ✓ | ✓ | ✓ |
| BFCL v3 (Patil et al., 2025) | ✓ | 15.7% | 39.7% | ✗ | ✓ | ✗ |
| BFCL v2 (Patil et al., 2025) | ✗ | 0.0% | 0.0% | ✗ | ✓ | ✗ |
| BFCL v1 (Patil et al., 2025) | ✗ | 0.0% | 0.0% | ✗ | ✓ | ✗ |
| ToolBench (Qin et al., 2024) | ✗ | 0.0% | 0.0% | ✓ | ✗ | ✗ |
| AnyToolBench (Du et al., 2024) | ✗ | 0.0% | 0.0% | ✓ | ✗ | ✗ |
| $\tau^2$-bench (Barres et al., 2025) | - | - | - | ✓ | ✗ | ✗ |
| $\tau$-bench (Yao et al., 2024) | - | - | - | ✓ | ✗ | ✗ |
| T-EVAL (Chen et al., 2024) | ✗ | 0.0% | 0.0% | ✓ | ✗ | ✗ |
| UltraTool (Huang et al., 2024a) | ✗ | 0.0% | 0.0% | ✓ | ✗ | ✗ |

dialogue, and §4.4 presents how frequent instruction transitions affect the LLM's ability to make correct decisions. Finally, in §4.5 we provide an empirical analysis of the errors that LLMs made in *WildToolBench*.

Table 2: WildToolBench Results.

| Models | Categorized by Task Type $g$ | | | | Categorized by Task Order $M$ | | | | Overall | |
|---|---|---|---|---|---|---|---|---|---|---|
| | $g_{single}$ | $g_{multi}$ | $g_{clarify}$ | $g_{chat}$ | 1 | 2 | 3 | 4 | Task | Session |
| *Proprietary General Models* | | | | | | | | | | |
| G Gemini-2.0-Thinking | 56.64 | 40.23 | **52.34** | **94.92** | **78.13** | **63.67** | 51.95 | **50.39** | **61.04** | **14.45** |
| G Gemini-2.5-Pro | 55.08 | 36.33 | 46.88 | 86.72 | 70.31 | 56.64 | 53.52 | 44.53 | 56.25 | 14.06 |
| A\ Claude-4-Sonnet | **60.16** | 43.75 | 41.80 | 80.47 | 71.09 | 57.81 | 52.73 | 44.53 | 56.54 | 12.50 |
| ⑤ o1 | 54.30 | 39.06 | 48.05 | 93.75 | 69.53 | 60.94 | **55.86** | 48.83 | 58.79 | 12.11 |
| ⑤ GPT-4o | **60.16** | 41.80 | 39.45 | 78.13 | 72.66 | 55.08 | 46.09 | 45.70 | 54.88 | 11.72 |
| ∅ Grok-4 | 59.38 | 41.41 | 33.59 | 66.02 | 63.67 | 52.34 | 42.97 | 41.41 | 50.10 | 10.16 |
| ⑤ GPT-5 | 46.09 | 34.38 | 31.64 | 84.38 | 62.11 | 50.00 | 45.31 | 39.06 | 49.12 | 5.86 |
| *Open-Source General Models* | | | | | | | | | | |
| ❄ GLM-4.5 | 57.81 | 40.63 | **44.53** | 81.25 | 70.70 | **60.16** | 50.78 | 42.58 | **56.05** | **12.11** |
| Ḱ Kimi-K2 | 54.30 | 33.98 | 39.84 | **86.72** | 68.75 | 57.03 | 48.83 | 40.23 | 53.71 | 10.55 |
| ❦ DeepSeek-R1 | 56.25 | **41.02** | 43.75 | 80.08 | 74.22 | 54.30 | 48.83 | **43.75** | 55.27 | 9.38 |
| ❦ DeepSeek-V3 | **58.98** | 38.67 | 33.59 | 79.30 | **75.39** | 53.91 | 41.02 | 40.23 | 52.64 | 9.38 |
| ❧ Qwen3-32B-Thinking | 53.52 | 28.91 | 37.11 | 80.86 | 62.50 | 52.73 | 46.48 | 38.67 | 50.10 | 7.81 |
| *Open-Source Specialized Models* | | | | | | | | | | |
| ☁ xLAM-2-70B | **64.45** | 36.72 | 28.91 | 64.84 | 64.06 | 51.56 | 42.58 | 36.72 | 48.73 | **7.81** |
| ▦ ToolACE2-8B | 62.11 | **37.89** | **33.98** | 84.38 | **72.27** | **59.38** | **46.88** | **39.84** | **54.59** | 7.42 |
| ⊗ Watt-8B | 61.72 | 28.13 | 22.66 | 78.13 | 68.75 | 47.27 | 39.06 | 35.55 | 47.66 | 4.69 |
| ⊞ Hammer2.1-7B | 40.23 | 21.88 | 30.47 | **94.92** | 61.72 | 46.88 | 40.63 | 38.28 | 46.88 | 4.69 |

## 4.1 OVERALL PERFORMANCE

We evaluate three categories of models, including **Proprietary General Models** (OpenAI, 2025; Anthropic, 2025; Mistral, 2024; Doubao, 2025; OpenAI, 2024b; Google, 2024; Seed, 2025), **Open-Source General Models** (Qwen, 2025; DeepSeek-AI et al., 2024; Yang et al., 2024; DeepSeek-AI et al., 2025), and Open-Source **Specialized Models** (Zeng et al., 2025; Liu et al., 2024b; Lin et al., 2024; Shi et al., 2024). We employ each model's native Function Call format to achieve optimal performance. Table 2 presents the overall performance of top-performing models. Full results on 57 models are provided in Appendix E.2.

In terms of overall performance, none of the mainstream LLMs achieve a session accuracy higher than 15%, and most models fall below 60% in task accuracy, highlighting the difficulty of *WildToolBench*. Proprietary LLMs generally outperform open-source ones, and reasoning-oriented models consistently surpass non-reasoning models. The best-performing open-source models, such as GLM4.5 and Kimi K2, achieve performance comparable to the top three proprietary models, while the remaining open-source models still lag considerably behind.

We further conducted a drill-down analysis of task accuracy along two dimensions: task type and task order. For task type, when the user's intention is casual chat or tool-free answering, most LLMs can reliably recognize the intention and respond appropriately. However, when the intention involves

clarification or eliciting task details through counter-questions, LLMs frequently misfire by executing a function call. Moreover, multi-step tool-use exhibits substantially lower accuracy than single-step tool invocation. For task order, within a dialogue, tasks appearing later exhibit greater dependence on preceding information, and model performance deteriorates accordingly.

## 4.2 LLMs Perform Pool on Tool Orchestration

Table 3: *WildToolBench* tool orchestration evaluation result.

| Models | Task Accuracy | | | | AP Rate | | | OP Rate | | |
|---|---|---|---|---|---|---|---|---|---|---|
| | $g_{\text{multi}}^P$ | $g_{\text{multi}}^S$ | $g_{\text{multi}}^{S+P}$ | Overall | $g_{\text{multi}}^S$ | $g_{\text{multi}}^{S+P}$ | Overall | $g_{\text{multi}}^P$ | $g_{\text{multi}}^{S+P}$ | Overall |
| *Proprietary General Models* | | | | | | | | | | |
| G Gemini-2.0-Thinking | 54.14 | 25.00 | 16.67 | 40.23 | 45.28 | 39.89 | 40.37 | **53.50** | 16.67 | 40.66 |
| G Gemini-2.5-pro | 49.04 | 25.00 | 14.29 | 36.33 | 47.17 | 39.15 | 39.87 | 43.31 | 11.90 | 32.37 |
| A\ Claude-4-Sonnet | **54.78** | 31.25 | **25.00** | **43.75** | **60.38** | 46.32 | **47.57** | 52.87 | **23.81** | **42.74** |
| ⑤ o1 | 50.96 | 12.50 | 21.43 | 39.06 | 35.85 | 37.50 | 37.35 | 50.32 | 20.24 | 39.83 |
| ⑤ GPT-4o | 53.50 | 31.25 | 21.43 | 41.80 | 41.51 | 45.40 | 45.06 | 51.59 | 21.43 | 41.08 |
| Ø Grok-4 | 54.14 | 18.75 | 21.43 | 41.41 | 41.51 | **46.51** | 46.06 | **53.50** | 21.43 | 42.32 |
| ⑤ GPT-5 | 43.31 | **37.50** | 16.67 | 34.38 | 49.06 | 38.42 | 39.36 | 42.68 | 13.10 | 32.37 |
| *Open-Source General Models* | | | | | | | | | | |
| ❄ GLM-4.5 | 51.59 | **31.25** | **21.43** | 40.63 | **67.92** | **48.90** | **50.59** | 49.68 | **20.24** | 39.42 |
| Ж Kimi-K2 | 45.86 | 12.50 | 15.48 | 33.98 | 52.83 | 34.93 | 36.52 | 43.95 | 15.48 | 34.02 |
| ❧ DeepSeek-R1 | **53.50** | 18.75 | **21.43** | **41.02** | 41.51 | 44.12 | 43.89 | **52.87** | **20.24** | **41.49** |
| ❧ DeepSeek-V3 | 52.87 | 25.00 | 14.29 | 38.67 | 43.40 | 32.54 | 33.50 | 51.59 | 14.29 | 38.59 |
| 🐦 Qwen3-32B-Thinking | 42.04 | 12.50 | 7.14 | 28.91 | 41.51 | 28.31 | 29.48 | 40.13 | 7.14 | 28.63 |
| *Open-Source Specialized Models* | | | | | | | | | | |
| ☁ xLAM-2-70B | **49.68** | 12.50 | 16.67 | 36.72 | 43.40 | **44.85** | **44.72** | 26.75 | 7.14 | 19.92 |
| ▦ ToolACE2-8B | 47.77 | **31.25** | **20.24** | **37.89** | **50.94** | 43.01 | 43.72 | 26.11 | **14.29** | 21.99 |
| ◎ Watt-8B | 44.59 | 6.25 | 1.19 | 28.13 | 22.64 | 21.87 | 21.94 | **44.59** | 1.19 | **29.46** |
| 🔨 Hammer2.1-7B | 33.12 | 12.50 | 2.38 | 21.88 | 24.53 | 13.24 | 14.24 | 31.85 | 2.38 | 21.58 |

We further analyzed whether LLMs can correctly orchestrate tool-call topologies to handle compositional tasks. We divided compositional tasks into three categories according to their required tool topology: sequential multi-step tool-use ($g_{\text{multi}}^S$), parallel multi-step tool-use ($g_{\text{multi}}^P$), and mixed tool-use combining both sequential and parallel structures ($g_{\text{multi}}^{S+P}$). As shown in Table 3, the highest task accuracy is merely 43.75%, falling to just 25% for $g_{\text{multi}}^{S+P}$ tasks, indicating that compositional tasks with multi-turn interactions remain a significant challenge for LLMs. Similarly, the peak optimal path (OP) rate reaches only 42.74%, suggesting that current LLMs have substantial room for improvement in tool execution efficiency. See full results of 57 LLMs in Appendix E.3. Specialized tool-use models perform significantly worse than general-purpose models, indicating limited generalization despite their intended focus. Claude-4-Sonnet shows a clear advantage in complex reasoning for tool orchestration, outperforming other proprietary models. The Gemini series reveals a strong bias, excelling in parallel but dropping sharply in mixed one. Among open-source models, GLM-4.5 excels in sequential and mixed tasks, even surpassing leading proprietary models. Furthermore, we observe that reasoning-enabled model variants outperform their non-reasoning counterparts within the same series, indicating that additional reasoning leads to better tool-call orchestration for compositional tasks. These results refute the conclusion in previous work Zhou et al. (2025) that a reasoning model does not outperform a non-reasoning model on tool-use, highlighting limitations in previous evaluations.

## 4.3 LLMs Struggle to Infer intention Across Dialogue

Figure 7 reports the accuracy of LLMs on three types of user tasks, in which user intention and information are partially hidden or omitted across multi-turn contexts. We find that long-range dependency tasks are the most challenging, with no model achieving accuracy above 50%. By contrast, tasks involving partial information or coreference are relatively easier. The results reveal clear specialization across models: reasoning models such as o1 and gemini-2.0-thinking excel at inferring omitted information and intent in partial information tasks, while Claude-4-Sonnet leads on coreferential reference tasks, indicating that no single model outperforms others across all aspects.

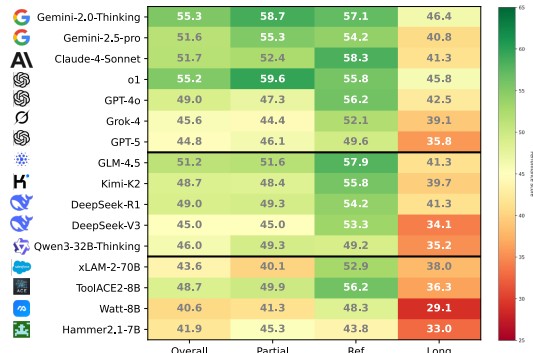

Figure 7: LLM's performance under different hidden information strategies.

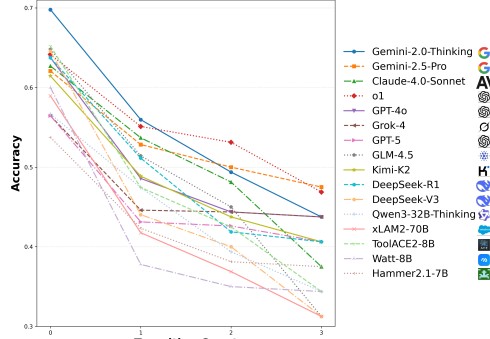

Figure 8: LLM's performance goes down as the instruction transition goes more frequently.

Long-range dependency tasks remain the weakest dimension overall, with scores clustered between 30 and 45, while also exhibiting the largest performance gap (17.3), making them a key differentiator among models. Mid-tier models demonstrate notable strengths despite lower overall averages; for instance, ToolACE2-8B (Ref 56.2) and GLM-4.5 (Ref 57.9) approach top-tier performance. Model capability generally correlates positively with model size, as illustrated by the Qwen2.5 series results in Table 6. In general, reasoning models demonstrate stronger capabilities in inferring hidden intent and retrieving omitted information within multi-turn contexts.

## 4.4 USER INSTRUCTIONS CHANGE, AND LLMS LAG BEHIND

To further investigate the impact of user instruction transitions on LLM decision-making, we analyzed the performance of all models on tasks in *WildToolBench* with varying transition frequencies. As stated in Section §3.1, we categorize the tasks into four types: $g_{single}$, $g_{multi}$, $g_{chat}$, and $g_{clarify}$, corresponding to tasks solvable with a single tool call, multi-step tool calls, direct question answering, and tasks requiring clarification, respectively. An instruction transition is defined as a change in task type between two consecutive tasks within a dialogue. Given that each scenario contains up to four tasks, at most three transitions can occur. As shown in Figure 8, across open-source and proprietary models, general-purpose and specialized models, as well as reasoning and non-reasoning models, task accuracy decreases as the number of transitions increases. In some cases, the drop reaches as much as **30%** in accuracy. Our analysis indicates two main factors underlying this trend. First, tasks with frequent transitions reflect more flexible and in-depth user demands (e.g., a task requiring clarification followed by a follow-up query for further information). Such tasks more closely resemble real user scenarios and are inherently more difficult. Second, LLMs exhibit self-conditioning (Sinha et al., 2025), whereby previous responses bias subsequent decisions. For example, if a model previously used a tool call, it tends to continue using tool calls; if it previously executed parallel tool calls, it is biased toward repeating them. This interference prevents the model from selecting the appropriate response. Essentially, this arises because the long conversational context dilutes the model's attention to the current task, as historical user and assistant messages accumulate. This problem is particularly pronounced when the current task requires recalling past interactions (as noted in Section §3.4), further exacerbating the interference from historical context.

## 4.5 ERROR ANALYSIS

Table 4 reveals that the primary challenge in LLM tool-use has shifted from syntactic correctness to semantic and logical reasoning. The data indicates two divergent failure philosophies: a "cautious" profile, exemplified by Gemini-2.0-Thinking, which prefers to refuse a task (24.56% Refusal rate) rather than risk an incorrect action (8.02% Wrong Name error), and an "eager" profile, seen in models like Grok-4, which minimizes refusals (3.72%) at the cost of a significantly higher propensity to select the wrong tool (24.07% Wrong Name error). Across the spectrum, "Wrong Name / Missing Info" and "Redundant Call" (23.06% in Gemini-2.0-Thinking) emerge as the most prevalent errors, highlighting systemic deficits in intent understanding and context management. This problem is particularly pronounced in specialized open-source models like xLAM-2-70B and Watt-8B, where "Wrong Name"

Table 4: Error Analysis in *WildToolBench*.

| Models | Action Errors | | | | | | Parameter Errors | | |
|---|---|---|---|---|---|---|---|---|---|
| | Refusal | Wrong Name Missing Info | Wrong Refusal | Redundant Call | Call Error | Early Termination | Param Type Error | Param Hallucination | Param Value Error |
| *Proprietary General Models* | | | | | | | | | |
| G Gemini-2.0-Thinking | 24.56% | 8.02% | 3.26% | 23.06% | 18.05% | 4.76% | 1.50% | 4.51% | 12.28% |
| G Gemini-2.5-Pro | 33.93% | 7.81% | 3.79% | 16.74% | 14.51% | 5.13% | 1.12% | 6.47% | 10.49% |
| A Claude-4-Sonnet | 9.44% | 19.55% | 11.24% | 16.40% | 12.13% | 6.52% | 1.80% | 8.09% | 14.83% |
| ⑤ o1 | 30.57% | 8.53% | 3.55% | 21.33% | 8.77% | 8.06% | 1.42% | 6.40% | 11.37% |
| ⑤ GPT-4o | 5.41% | 21.65% | 12.12% | 14.50% | 11.47% | 7.58% | 3.46% | 9.96% | 13.85% |
| Ø Grok-4 | 3.72% | 24.07% | 17.03% | 17.81% | 10.18% | 5.68% | 2.94% | 6.46% | 12.13% |
| ⑤ GPT-5 | 15.93% | 13.05% | 6.91% | 31.67% | 10.17% | 3.65% | 1.34% | 10.75% | 6.53% |
| *Open-Source General Models* | | | | | | | | | |
| ✴ GLM-4.5 | 10.89% | 19.33% | 10.67% | 18.89% | 15.33% | 6.00% | 1.33% | 4.67% | 12.89% |
| Ӄ Kimi-K2 | 21.31% | 13.50% | 7.17% | 16.24% | 11.60% | 6.54% | 2.74% | 6.75% | 14.14% |
| 🐋 DeepSeek-R1 | 13.54% | 14.41% | 11.14% | 20.96% | 11.79% | 6.33% | 1.53% | 8.52% | 11.79% |
| 🐋 DeepSeek-V3 | 10.52% | 21.65% | 10.93% | 15.88% | 16.49% | 5.15% | 1.65% | 7.42% | 10.31% |
| 🦅 Qwen3-32B-Thinking | 9.20% | 20.35% | 9.20% | 19.18% | 19.18% | 4.31% | 1.96% | 7.83% | 8.81% |
| *Open-Source Specialized Models* | | | | | | | | | |
| ☁ xLAM-2-70B | 6.48% | 30.67% | 17.14% | 4.38% | 16.19% | 5.71% | 1.71% | 5.14% | 12.57% |
| ▓ ToolACE2-8B | 10.11% | 28.60% | 8.60% | 6.67% | 18.28% | 6.02% | 3.23% | 3.66% | 14.84% |
| ◎ Watt-8B | 5.97% | 30.97% | 10.45% | 7.09% | 23.13% | 4.29% | 1.87% | 5.78% | 10.45% |
| 🔨 Hammer2.1-7B | 38.24% | 15.81% | 2.39% | 12.68% | 15.26% | 1.84% | 3.49% | 1.47% | 8.82% |

errors exceed 30%, suggesting that specialization can lead to brittleness. Conversely, parameter-level errors such as "Param Type Error" or "Param Hallucination" are consistently lower across all models. This suggests that the frontier of agentic AI development now lies in improving higher-order planning and reasoning rather than basic syntactic generation. The prevalence of "Redundant Call" errors reveals a widespread deficiency in long-range planning for most capable models, indicating that they struggle with context management over time. However, the deceptively low rates of this error in some specialized models can be misleading, as this "pseudo-capability" often masks a more fundamental failure to initiate tasks correctly, evidenced by catastrophic "Wrong Name" error rates.

## 5   LIMITATIONS

*WildToolBench* uses human annotations to ensure data quality, diversity, and alignment with real user behaviour distribution. However, this limits the scaling potential of the data size. What's more, the dual objectives of maintaining data quality and traversing all policy transition types concurrently limit the feasible length of tasks. Despite this, experimental results reveal significant trends in model performance, leading to robust conclusions on the gap in current LLMs' tool-use ability. We are also working on combining human-annotated rubrics with a fully automated synthetic environment scaling pipeline for both training and evaluation of the Agentic Model, which is the foundation for the next scaling trend in the AI era.

## 6   CONCLUSION

*WildToolBench*, grounded in real user behavior patterns, identifies three major challenges for LLMs performing multi-turn, multi-step tool-use: compositional instructions, hidden intent, and instruction transitions. Unlike prior evaluations that focus solely on increasing the complexity of tool-call procedures, *WildToolBench* emphasizes assessing LLM tool-use capabilities in the context of realistic user scenarios. Benchmarking nearly all mainstream models, *WildToolBench* reveals a fundamental limitation in current LLM development: for effective tool-use, a model cannot merely function as a tool executor; it must also possess the capacity to understand users. This capability depends on deeper foundational skills of large models, including instruction following, long-context comprehension, and theory of mind—essential abilities for future agentic models. Beyond serving as a leaderboard, *WildToolBench* provides structured rubrics that guide model developers in interpreting user behaviors from multiple perspectives, facilitating more effective model iteration.

## 7 REPRODUCIBILITY STATEMENT

*WildToolBench* provides all the datasets, evaluation scripts, and all 57 LLMs evaluated trajectories to support 100% reproducibility. See these materials in the Github Repository.

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

## A  THE USE OF LARGE LANGUAGE MODELS (LLMS)

For the paper writing, we employed LLMs solely for grammatical correction at the writing stage. The LLM itself did not contribute to experimental design, idea development, or manuscript writing.

Other uses of LLMs (such as benchmark construction) have been clearly stated in §3.2 and §C.

## B  BENCHMARK COMPARISON

Since the representative benchmarks we compiled span multiple time periods, not all models reported results for every benchmark in their original papers. Therefore, we collected evaluation results from multiple sources. Figure 1 primarily demonstrates that previous LLM tool benchmarks have tended toward saturation, while WildToolBench remains challenging. The information we compiled is mainly drawn from official reports of GLM4.5, Kimi K2, GPT5, and BFCL leaderboard[1], as well as a report by an independent third-party organization, Artificial Analysis[2]. The details are shown in Table 5.

Table 5: Performance Comparison across Different Benchmarks

| Models | WildToolBench | BFCL-v4 | BFCL-v3 | BFCL-v2 | $\tau$-Bench | $\tau^2$-Bench (telecom) |
|---|---|---|---|---|---|---|
| G Gemini2.5 pro | **14.1** | 48.6 | 25.0 | 64.0 | 62.5 | 54.1 |
| ⑤ GPT4o | 11.7 | 64.7 | 42.5 | 70.5 | 51.6 | 25.1 |
| A\ Claude-Sonnet 4 | 12.5 | **71.7** | 54.8 | 81.1 | 70.3 | 45.2 |
| A\ Claude-Opus 4 | 9.4 | 69.6 | 57.9 | **81.5** | **70.5** | 57.0 |
| ⑤ GPT5 | 5.9 | 71.1 | 28.5 | 58.3 | 72.3 | **96.7** |
| Ø Grok 4 | 10.2 | 68.9 | 36.1 | 74.4 | 67.5 | 47.9 |
| ❋ GLM4.5 | **12.1** | 64.9 | **65.6** | 81.7 | 70.1 | 24.6 |
| Ӄ K2 | 10.6 | 42.1 | 48.8 | 80.8 | 62.6 | 65.8 |
| ❧ DeepSeek R1 | 9.4 | 32.0 | 44.5 | 80.9 | 58.7 | 36.5 |
| ❧ DeepSeek V3 | 9.0 | 27.4 | 33.0 | 79.9 | 57.6 | 32.5 |

## C  DATA CURATION

Figure 3 gives an overall preview of the data curation for *WildToolBench*. First, we analyzed a large collection of real user logs to extract suitable seed scenarios and to summarize user behavior patterns. Second, we build our toolset by leveraging tool descriptions from public APIs[3], following the approach introduced by ToolAlpaca. This GitHub repository is continuously updated and now contains more than 1400 tool lists, but to stay consistent with ToolAlpaca, we use 400 of these tool lists, covering around 1600 APIs in total. Then, we selected a corresponding tool subset for each seed scenario. In particular, we enumerated all possible simple and complex parameter types (String, Integer, Float, Boolean, Enum, Array, Object, Nested) to enhance the diversity and complexity of tool parameters. Third, five human experts specializing in LLM agents inspected and refined these tool sets, mainly by correcting unreasonable tool combinations and parameter specifications, thereby improving the logical coherence and interoperability of tools. This process yielded 256 realistic scenarios, and for each scenario, we get a diverse and reasonable tool subset.

After obtaining the scenarios, we prompted a User Agent to generate initial first-round user tasks based on the scenario and tool subset. Based on the four task types defined in this paper ($g_{single}$, $g_{multi}$, $g_{clarify}$, $g_{chat}$), we used controlled generation to produce the first-round tasks for each type. To enhance diversity, we varied across five dimensions: sentence structure, linguistic style, task background, task length, and task difficulty. We then used the three omission types defined in Challenge 2 (User Hidden Intent), including Partial Information, Coreferential Reference, and Long-Range Dependence, together with real user questions as few-shot examples, to guide the User Agent in generating the subsequent three tasks. For each step, multiple candidate tasks were generated, from which human experts selected the highest-quality ones and refined them to better match human distributions, resulting in the final user tasks.

---

[1]https://gorilla.cs.berkeley.edu/leaderboard.html

[2]https://artificialanalysis.ai/

[3]https://github.com/public-apis/public-apis

Once the expert-refined user tasks were obtained, the Assistant Agent executed the Agent Loop for tool calls until producing a summary. Each tool call in the trajectory was then automatically checked for issues such as function hallucination, parameter hallucination, type errors, and redundant calls. Subsequently, five human experts inspected the full trajectory, corrected errors (e.g., in tool planning or parameter values), and annotated tool-call dependencies (used to construct DAGs for calculating optimal path rates in Challenge 1). This process was repeated until all task trajectories were generated.

Finally, to ensure data quality, we conducted multiple rounds of discussion-based optimization. Several human experts randomly sampled 20% of the data, annotated potential errors, and initiated a review session where annotators collectively resolved issues. This process was repeated with different pairs of experts and different 20% samples each round, continuing until the detected error rate dropped to zero and every data point had been checked at least once with no conflict. After four such iterations, the data quality improved from 62%, 78%, 86%, and 94% to a final 100%, yielding the completed *WildToolBench*. 9 human experts took one month to finish the whole data curation process.

## D  Details of Tool Orchestration Evaluation

Algorithm 1 shows the pseudo code for enumerating all possible tool orchestration paths. The main design idea of this algorithm is to enumerate all possible tool execution paths in a directed acyclic graph (DAG) using depth-first search with backtracking. At each step, it identifies the set of nodes with zero indegree, generates all non-empty subsets to simulate parallel execution, appends the selected nodes to the current path, and updates the indegrees of their successors. The algorithm recursively explores all possible paths and finally classifies them into optimal and suboptimal sets based on path length, systematically accounting for both serial and parallel execution combinations (mixed multi-tool).

## E  Complete Experimental Results

### E.1  Hyperparameter Settings

To further enhance the reproducibility of our dataset, we hereby introduce the hyperparameter settings used during model inference. Specifically:

For **Proprietary Models**, we adopted the default hyperparameters from the official website without making any changes to hyperparameters such as `temperature`, `top-p`, and `top-k`.

For **Open-Source Models**, if an official API is available, we utilize it with its default hyperparameters. Otherwise, the model is deployed via the Hugging Face library, where tool-calling functionality is implemented according to its chat template. For generations, we use the model.generate method with its default hyperparameters, setting only max_new_tokens to 512. The version of Hugging Face used was 4.51.0, and no other modifications were made.

### E.2  *WildToolBench* Full Results

We provide all the benchmarking results of 57 models as shown in Table 6, including 16 Proprietary General Models, 30 Open-Source General Models, and 11 Open-Source Specialized Models trained for tool-use.

### E.3  *WildToolBench* Full Tool Orchestration Result

We provide all the detailed results on tool orchestration of 57 models as shown in Table 7, including 16 Proprietary General Models, 30 Open-Source General Models, and 11 Open-Source Specialized Models trained for tool-use.

### E.4  *WildToolBench* Full Error Analysis

We provide all the detailed error analysis results of 57 models as shown in Table 8, including 16 Proprietary General Models, 30 Open-Source General Models, and 11 Open-Source Specialized Models trained for tool-use.

Table 6: WildToolBench Full Results.

| Models | Categorized by Task Type $g$ | | | | Categorized by Task Order $M$ | | | | Overall | |
|---|---|---|---|---|---|---|---|---|---|---|
| | $g_{single}$ | $g_{multi}$ | $g_{clarify}$ | $g_{chat}$ | 1 | 2 | 3 | 4 | Task | Session |
| *Proprietary General Models* | | | | | | | | | | |
| G Gemini-2.0-Thinking | 56.64 | 40.23 | **52.34** | **94.92** | **78.13** | **63.67** | 51.95 | **50.39** | **61.04** | **14.45** |
| G Gemini-2.5-Pro | 55.08 | 36.33 | 46.88 | 86.72 | 70.31 | 56.64 | 53.52 | 44.53 | 56.25 | 14.06 |
| A\ Claude-4-Sonnet | 60.16 | 43.75 | 41.80 | 80.47 | 71.09 | 57.81 | 52.73 | 44.53 | 56.54 | 12.50 |
| o1 | 54.30 | 39.06 | 48.05 | 93.75 | 69.53 | 60.94 | **55.86** | 48.83 | 58.79 | 12.11 |
| GPT-4o | 60.16 | 41.80 | 39.45 | 78.13 | 72.66 | 55.08 | 46.09 | 45.70 | 54.88 | 11.72 |
| A\ Claude-3.7-Sonnet | 57.81 | 39.06 | 41.41 | 63.28 | 60.55 | 50.00 | 48.05 | 42.97 | 50.39 | 11.33 |
| o3 | **61.72** | 39.45 | 44.92 | 81.64 | 73.83 | 60.94 | 49.22 | 43.75 | 56.93 | 10.16 |
| Ø Grok-4 | 59.38 | 41.41 | 33.59 | 66.02 | 63.67 | 52.34 | 42.97 | 41.41 | 50.10 | 10.16 |
| A\ Claude-4.1-Opus | 55.86 | 39.84 | 41.80 | 82.42 | 69.92 | 55.08 | 50.39 | 44.53 | 54.98 | 9.38 |
| GPT-4.1 | 57.42 | **44.14** | 34.38 | 81.25 | 69.53 | 58.20 | 46.88 | 42.58 | 54.30 | 8.98 |
| M Mistral-Large | 56.25 | 36.33 | 37.50 | 68.75 | 67.58 | 48.83 | 44.14 | 38.28 | 49.71 | 7.03 |
| D Doubao-1.6 | 55.86 | 40.23 | 31.25 | 63.28 | 69.14 | 48.83 | 40.23 | 32.42 | 47.66 | 7.03 |
| D Doubao-1.5-Thinking | 60.16 | 22.66 | 26.95 | 75.39 | 65.63 | 47.66 | 40.23 | 31.64 | 46.29 | 6.64 |
| GPT-5 | 46.09 | 34.38 | 31.64 | 84.38 | 62.11 | 50.00 | 45.31 | 39.06 | 49.12 | 5.86 |
| D Doubao-1.6-Thinking | 57.42 | 34.38 | 18.75 | 47.27 | 57.03 | 39.06 | 31.25 | 30.47 | 39.45 | 3.13 |
| D Doubao-1.5 | 58.59 | 24.61 | 9.38 | 34.38 | 39.45 | 28.91 | 29.30 | 29.30 | 31.74 | 0.78 |
| *Open-Source General Models* | | | | | | | | | | |
| ☀ GLM-4.5 | 57.81 | 40.63 | **44.53** | 81.25 | 70.70 | **60.16** | **50.78** | 42.58 | **56.05** | **12.11** |
| K Kimi-K2 | 54.30 | 33.98 | 39.84 | 86.72 | 68.75 | 57.03 | 48.83 | 40.23 | 53.71 | 10.55 |
| Qwen3-30B-A3B | 48.05 | 28.13 | 41.41 | 89.06 | 69.92 | 51.56 | 48.05 | 37.11 | 51.66 | 9.77 |
| Qwen3-14B-Thinking | 56.64 | 30.47 | 37.11 | 88.67 | 69.53 | 54.30 | 50.39 | 38.67 | 53.22 | 9.38 |
| DeepSeek-R1 | 56.25 | **41.02** | 43.75 | 80.08 | 74.22 | 54.30 | 48.83 | **43.75** | 55.27 | 9.38 |
| DeepSeek-V3 | 58.98 | 38.67 | 33.59 | 79.30 | **75.39** | 53.91 | 41.02 | 40.23 | 52.64 | 9.38 |
| Qwen3-8B-Thinking | 56.64 | 33.59 | 39.84 | 87.11 | 73.05 | 55.08 | 48.83 | 40.23 | 54.30 | 8.98 |
| DeepSeek-V3.1 | 44.92 | 40.63 | 33.98 | 81.25 | 61.33 | 51.56 | 45.70 | 42.19 | 50.20 | 8.98 |
| Qwen3-32B | 57.81 | 33.20 | 39.84 | 79.69 | 69.53 | 54.30 | 46.48 | 40.23 | 52.64 | 8.59 |
| Qwen2.5-14B-Instruct | 50.39 | 27.34 | 32.81 | 83.20 | 66.41 | 48.44 | 40.63 | 38.28 | 48.44 | 7.81 |
| Qwen3-14B | 56.25 | 31.25 | 39.84 | 89.06 | 71.09 | 54.30 | 50.39 | 40.63 | 54.10 | 7.81 |
| Qwen3-32B-Thinking | 53.52 | 28.91 | 37.11 | 80.86 | 62.50 | 52.73 | 46.48 | 38.67 | 50.10 | 7.81 |
| Qwen3-4B-Thinking | **60.16** | 28.91 | 38.67 | 87.89 | 68.75 | 59.38 | 44.14 | 43.36 | 53.91 | 7.81 |
| Qwen2.5-72B-Instruct | 58.98 | 32.03 | 34.38 | 82.81 | 70.70 | 50.00 | 46.48 | 41.02 | 52.05 | 6.25 |
| Qwen3-8B | **60.16** | 26.17 | 26.95 | 79.30 | 66.02 | 48.44 | 39.45 | 38.67 | 48.14 | 6.25 |
| Qwen3-30B-A3B-Thinking | 50.39 | 24.61 | 38.67 | 89.45 | 71.88 | 49.22 | 44.14 | 37.89 | 50.78 | 6.25 |
| Qwen3-1.7B-Thinking | 49.22 | 24.61 | 30.08 | 84.38 | 68.75 | 46.09 | 41.80 | 31.64 | 47.07 | 6.25 |
| Qwen2.5-32B-Instruct | 53.91 | 38.67 | 36.72 | 82.81 | 69.14 | 54.30 | 48.83 | 39.84 | 53.03 | 5.86 |
| Qwen3-4B | 51.17 | 22.66 | 35.94 | 86.33 | 69.14 | 49.22 | 39.06 | 38.67 | 49.02 | 5.86 |
| Qwen2.5-7B-Instruct | 52.73 | 25.39 | 28.91 | 73.05 | 64.06 | 45.31 | 37.50 | 33.20 | 45.02 | 4.30 |
| Qwen2.5-3B-Instruct | 48.83 | 18.36 | 20.70 | 73.83 | 55.86 | 41.41 | 33.59 | 30.86 | 40.43 | 4.30 |
| Qwen3-1.7B | 48.05 | 19.53 | 26.95 | 81.64 | 67.58 | 44.14 | 33.98 | 30.47 | 44.04 | 4.30 |
| Qwen2.5-1.5B-Instruct | 36.72 | 13.28 | 21.48 | **90.23** | 60.16 | 39.06 | 31.64 | 30.86 | 40.43 | 3.91 |
| Qwen3-0.6B-Thinking | 44.92 | 16.02 | 23.05 | 87.11 | 66.02 | 40.23 | 35.55 | 29.30 | 42.77 | 3.91 |
| Qwen3-0.6B | 46.09 | 7.81 | 12.11 | 78.13 | 53.91 | 34.38 | 26.56 | 29.30 | 36.04 | 3.52 |
| Qwen2.5-0.5B-Instruct | 23.83 | 4.69 | 15.23 | 82.03 | 43.75 | 29.69 | 26.95 | 25.39 | 31.45 | 0.78 |
| ∞ Llama-3.3-3B-Instruct | 0.00 | 0.00 | 0.39 | 64.84 | 21.88 | 17.19 | 14.45 | 11.72 | 16.31 | 0.39 |
| ∞ Llama-3.3-1B-Instruct | 0.00 | 0.00 | 0.39 | 61.33 | 17.58 | 18.75 | 14.06 | 11.33 | 15.43 | 0.39 |
| ∞ Llama-3.3-70B-Instruct | 3.52 | 0.00 | 0.00 | 52.73 | 17.97 | 14.06 | 12.89 | 11.33 | 14.06 | 0.00 |
| ∞ Llama-3.3-8B-Instruct | 0.39 | 0.00 | 0.39 | 69.14 | 25.39 | 19.14 | 14.06 | 11.33 | 17.48 | 0.00 |
| *Open-Source Specialized Models* | | | | | | | | | | |
| xLAM-2-32B | 60.94 | 34.77 | **38.67** | 69.92 | 69.53 | 52.73 | 43.36 | 38.67 | 51.07 | **8.20** |
| xLAM-2-70B | **64.45** | 36.72 | 28.91 | 64.84 | 64.06 | 51.56 | 42.58 | 36.72 | 48.73 | 7.81 |
| ToolACE2-8B | 62.11 | **37.89** | 33.98 | 84.38 | **72.27** | 59.38 | 46.88 | 39.84 | **54.59** | 7.42 |
| xLAM-2-8B | 62.11 | 29.30 | 24.22 | 54.30 | 52.73 | 44.14 | 39.06 | 33.98 | 42.48 | 5.08 |
| Watt-8B | 61.72 | 28.13 | 22.66 | 78.13 | 68.75 | 47.27 | 39.06 | 35.55 | 47.66 | 4.69 |
| Hammer2.1-7B | 40.23 | 21.88 | 30.47 | **94.92** | 61.72 | 46.88 | 40.63 | 38.28 | 46.88 | 4.69 |
| xLAM-2-1B | 53.13 | 17.19 | 15.23 | 65.23 | 50.39 | 40.63 | 33.98 | 25.78 | 37.70 | 2.34 |
| xLAM-2-3B | 52.34 | 23.44 | 16.02 | 57.03 | 42.97 | 43.36 | 34.77 | 27.73 | 37.21 | 1.17 |
| Hammer2.1-3B | 32.81 | 18.36 | 11.33 | 91.02 | 34.77 | 46.48 | 40.63 | 31.64 | 38.38 | 0.39 |
| Hammer2.1-1.5B | 4.69 | 1.56 | 1.17 | **96.09** | 26.17 | 26.17 | 26.56 | 24.61 | 25.88 | 0.00 |
| Hammer2.1-0.5B | 9.77 | 2.34 | 3.13 | 86.72 | 25.39 | 29.69 | 25.39 | 21.48 | 25.49 | 0.00 |

Table 7: *WildToolBench* Tool Orchestration Result.

| Models | Task Accuracy | | | | AP Rate | | | OP Rate | | |
|---|---|---|---|---|---|---|---|---|---|---|
| | $g^P_{\text{multi}}$ | $g^S_{\text{multi}}$ | $g^{S+P}_{\text{multi}}$ | Overall | $g^S_{\text{multi}}$ | $g^{S+P}_{\text{multi}}$ | Overall | $g^P_{\text{multi}}$ | $g^{S+P}_{\text{multi}}$ | Overall |
| *Proprietary General Models* | | | | | | | | | | |
| G Gemini-2.0-Thinking | 54.14 | 25.00 | 16.67 | 40.23 | 45.28 | 39.89 | 40.37 | 53.50 | 16.67 | 40.66 |
| G Gemini-2.5-Pro | 49.04 | 25.00 | 14.29 | 36.33 | 47.17 | 39.15 | 39.87 | 43.31 | 11.90 | 32.37 |
| A\ Claude-3.7-Sonnet | 43.95 | **62.50** | **25.00** | 39.06 | **86.79** | **61.40** | **63.65** | 0.00 | 0.00 | 0.00 |
| A\ Claude-4-Sonnet | 54.78 | 31.25 | **25.00** | 43.75 | 60.38 | 46.32 | 47.57 | 52.87 | **23.81** | 42.74 |
| A\ Claude-4.1-Opus | 50.96 | 43.75 | 17.86 | 39.84 | 62.26 | 48.35 | 49.58 | 50.32 | 16.67 | 38.59 |
| ⑤ o1 | 50.96 | 12.50 | 21.43 | 39.06 | 35.85 | 37.50 | 37.35 | 50.32 | 20.24 | 39.83 |
| ⑤ o3 | 48.41 | 31.25 | 23.81 | 39.45 | 66.04 | 54.60 | 55.61 | 0.64 | 0.00 | 0.41 |
| ⑤ o4-mini | 39.49 | 31.25 | 16.67 | 31.64 | 52.83 | 37.68 | 39.03 | 0.00 | 0.00 | 0.00 |
| ⑤ GPT-4o | 53.50 | 31.25 | 21.43 | 41.80 | 41.51 | 45.40 | 45.06 | 51.59 | 21.43 | 41.08 |
| ⑤ GPT-4.1 | **58.60** | 25.00 | 20.24 | **44.14** | 49.06 | 45.77 | 46.06 | **56.69** | 19.05 | **43.57** |
| ⑤ GPT-5 | 43.31 | 37.50 | 16.67 | 34.38 | 49.06 | 38.42 | 39.36 | 42.68 | 13.10 | 32.37 |
| Ø Grok-4 | 54.14 | 18.75 | 21.43 | 41.41 | 41.51 | 46.51 | 46.06 | 53.50 | 21.43 | 42.32 |
| M Mistral-Large | 47.77 | 25.00 | 16.67 | 36.33 | 45.28 | 40.44 | 40.87 | 45.86 | 15.48 | 35.27 |
| ◐ Doubao-1.5 | 35.03 | 12.50 | 7.14 | 24.61 | 37.74 | 29.41 | 30.15 | 9.55 | 1.19 | 6.64 |
| ◐ Doubao-1.5-Thinking | 31.21 | 18.75 | 7.14 | 22.66 | 56.60 | 23.35 | 26.30 | 28.03 | 7.14 | 20.75 |
| ◐ Doubao-1.6 | 50.96 | 25.00 | 22.62 | 40.23 | 50.94 | 47.79 | 48.07 | 50.96 | 22.62 | 41.08 |
| ◐ Doubao-1.6-Thinking | 46.50 | 12.50 | 15.48 | 34.38 | 52.83 | 39.34 | 40.54 | 43.95 | 15.48 | 34.02 |
| *Open-Source General Models* | | | | | | | | | | |
| ➤ xLAM-2-70B | **49.68** | 12.50 | 16.67 | 36.72 | 43.40 | **44.85** | **44.72** | 26.75 | 7.14 | 19.92 |
| ➤ xLAM-2-32B | 45.86 | 25.00 | 15.48 | 34.77 | **58.49** | 40.26 | 41.88 | 25.48 | 5.95 | 18.67 |
| ➤ xLAM-2-8B | 40.76 | 25.00 | 8.33 | 29.30 | 43.40 | 27.57 | 28.98 | 26.75 | 3.57 | 18.67 |
| ➤ xLAM-2-3B | 33.12 | 12.50 | 7.14 | 23.44 | 24.53 | 23.35 | 23.45 | 15.92 | 3.57 | 11.62 |
| ➤ xLAM-2-1B | 27.39 | 0.00 | 1.19 | 17.19 | 22.64 | 17.46 | 17.92 | 10.83 | 0.00 | 7.05 |
| ▦ ToolACE2-8B | 47.77 | 31.25 | **20.24** | **37.89** | 50.94 | 43.01 | 43.72 | 26.11 | **14.29** | 21.99 |
| ◪ Watt-8B | 44.59 | 6.25 | 1.19 | 28.13 | 22.64 | 21.87 | 21.94 | **44.59** | 1.19 | **29.46** |
| ✦ Hammer2.1-7B | 33.12 | 12.50 | 2.38 | 21.88 | 24.53 | 13.24 | 14.24 | 31.85 | 2.38 | 21.58 |
| ✦ Hammer2.1-3B | 24.84 | 12.50 | 7.14 | 18.36 | 32.08 | 18.01 | 19.26 | 24.20 | 7.14 | 18.26 |
| ✦ Hammer2.1-1.5B | 2.55 | 0.00 | 0.00 | 1.56 | 1.89 | 1.29 | 1.34 | 2.55 | 0.00 | 1.66 |
| ✦ Hammer2.1-0.5B | 3.18 | 0.00 | 1.19 | 2.34 | 9.43 | 10.48 | 10.39 | 3.18 | 0.00 | 2.07 |
| *Open-Source Specialized Models* | | | | | | | | | | |
| ∞ Llama-3.3-70B-Instruct | 0.00 | 0.00 | 0.00 | 0.00 | 15.09 | 2.02 | 3.18 | 0.00 | 0.00 | 0.00 |
| ∞ Llama-3.3-8B-Instruct | 0.00 | 0.00 | 0.00 | 0.00 | 3.77 | 0.18 | 0.50 | 0.00 | 0.00 | 0.00 |
| ∞ Llama-3.3-3B-Instruct | 0.00 | 0.00 | 0.00 | 0.00 | 3.77 | 0.18 | 0.50 | 0.00 | 0.00 | 0.00 |
| ∞ Llama-3.3-1B-Instruct | 0.00 | 0.00 | 0.00 | 0.00 | 3.77 | 0.18 | 0.50 | 0.00 | 0.00 | 0.00 |
| ◈ Qwen2.5-72B-Instruct | 44.59 | 25.00 | 9.52 | 32.03 | 41.51 | 30.33 | 31.32 | 42.04 | 7.14 | 29.88 |
| ◈ Qwen2.5-32B-Instruct | 52.87 | **43.75** | 10.71 | 38.67 | 56.60 | 24.08 | 26.97 | 52.23 | 10.71 | 37.76 |
| ◈ Qwen2.5-14B-Instruct | 39.49 | 18.75 | 5.95 | 27.34 | 26.42 | 19.67 | 20.27 | 36.31 | 4.76 | 25.31 |
| ◈ Qwen2.5-7B-Instruct | 38.22 | 6.25 | 4.76 | 25.39 | 28.30 | 26.29 | 26.47 | 33.12 | 2.38 | 22.41 |
| ◈ Qwen2.5-3B-Instruct | 28.66 | 6.25 | 1.19 | 18.36 | 15.09 | 12.13 | 12.40 | 27.39 | 0.00 | 17.84 |
| ◈ Qwen2.5-1.5B-Instruct | 21.66 | 0.00 | 0.00 | 13.28 | 9.43 | 4.96 | 5.36 | 21.66 | 0.00 | 14.11 |
| ◈ Qwen2.5-0.5B-Instruct | 7.64 | 0.00 | 0.00 | 4.69 | 11.32 | 3.68 | 4.36 | 7.64 | 0.00 | 4.98 |
| ◈ Qwen3-30B-A3B | 42.04 | 6.25 | 5.95 | 28.13 | 26.42 | 25.00 | 25.13 | 40.76 | 5.95 | 28.63 |
| ◈ Qwen3-32B | 46.50 | 12.50 | 11.90 | 33.20 | 47.17 | 30.70 | 32.16 | 43.95 | 9.52 | 31.95 |
| ◈ Qwen3-14B | 44.59 | 25.00 | 7.14 | 31.25 | 41.51 | 28.86 | 29.98 | 44.59 | 7.14 | 31.54 |
| ◈ Qwen3-8B | 38.85 | 0.00 | 7.14 | 26.17 | 15.09 | 25.18 | 24.29 | 38.22 | 5.95 | 26.97 |
| ◈ Qwen3-4B | 35.67 | 6.25 | 1.19 | 22.66 | 28.30 | 13.42 | 14.74 | 34.39 | 1.19 | 22.82 |
| ◈ Qwen3-1.7B | 31.85 | 0.00 | 0.00 | 19.53 | 18.87 | 16.91 | 17.09 | 31.85 | 0.00 | 20.75 |
| ◈ Qwen3-0.6B | 12.10 | 6.25 | 0.00 | 7.81 | 22.64 | 9.19 | 10.39 | 11.47 | 0.00 | 7.47 |
| ◈ Qwen3-30B-A3B-Thinking | 36.94 | 6.25 | 4.76 | 24.61 | 18.87 | 22.43 | 22.11 | 35.67 | 3.57 | 24.48 |
| ◈ Qwen3-32B-Thinking | 42.04 | 12.50 | 7.14 | 28.91 | 41.51 | 28.31 | 29.48 | 40.13 | 7.14 | 28.63 |
| ◈ Qwen3-14B-Thinking | 43.95 | 25.00 | 5.95 | 30.47 | 45.28 | 31.62 | 32.83 | 43.95 | 5.95 | 30.71 |
| ◈ Qwen3-8B-Thinking | 47.13 | 12.50 | 11.90 | 33.59 | 28.30 | 31.99 | 31.66 | 47.13 | 10.71 | 34.44 |
| ◈ Qwen3-4B-Thinking | 42.68 | 18.75 | 4.76 | 28.91 | 39.62 | 23.16 | 24.62 | 42.68 | 4.76 | 29.46 |
| ◈ Qwen3-1.7B-Thinking | 38.85 | 6.25 | 1.19 | 24.61 | 26.42 | 11.40 | 12.73 | 38.85 | 1.19 | 25.73 |
| ◈ Qwen3-0.6B-Thinking | 26.11 | 0.00 | 0.00 | 16.02 | 18.87 | 9.74 | 10.55 | 25.48 | 0.00 | 16.60 |
| ✴ GLM-4.5 | 51.59 | 31.25 | **21.43** | 40.63 | **67.92** | **48.90** | **50.59** | 49.68 | **20.24** | 39.42 |
| Ϗ Kimi-K2 | 45.86 | 12.50 | 15.48 | 33.98 | 52.83 | 34.93 | 36.52 | 43.95 | 15.48 | 34.02 |
| ✆ DeepSeek-R1 | **53.50** | 18.75 | **21.43** | **41.02** | 41.51 | 44.12 | 43.89 | **52.87** | **20.24** | **41.49** |
| ✆ DeepSeek-V3 | 52.87 | 25.00 | 14.29 | 38.67 | 43.40 | 32.54 | 33.50 | 51.59 | 14.29 | 38.59 |
| ✆ DeepSeek-V3.1 | **53.50** | 25.00 | 19.05 | 40.63 | 52.83 | 37.68 | 39.03 | 47.77 | 14.29 | 36.10 |

Table 8: *WildToolBench* Full Error Distribution Analysis.

| Models | Action Errors | | | | | | Parameter Errors | | |
|---|---|---|---|---|---|---|---|---|---|
| | Refusal | Wrong Name Missing Info | Wrong Refusal | Redundant Call | Call Error | Early Termination | Param Type Error | Param Hallucination | Param Value Error |
| *Proprietary General Models* | | | | | | | | | |
| G Gemini-2.0-Thinking | 24.56% | 8.02% | 3.26% | 23.06% | 18.05% | 4.76% | 1.50% | 4.51% | 12.28% |
| G Gemini-2.5-Pro | 33.93% | 7.81% | 3.79% | 16.74% | 14.51% | 5.13% | 1.12% | 6.47% | 10.49% |
| A\ Claude-3.7-Sonnet | 11.02% | 16.73% | 17.91% | 16.34% | 16.54% | 1.57% | 1.57% | 6.30% | 12.01% |
| A\ Claude-4-Sonnet | 9.44% | 19.55% | 11.24% | 16.40% | 12.13% | 6.52% | 1.57% | 8.31% | 14.83% |
| A\ Claude-4.1-Opus | 15.18% | 17.79% | 9.76% | 18.22% | 12.15% | 3.25% | 2.17% | 8.46% | 13.02% |
| ⑤ o1 | 30.57% | 8.53% | 3.55% | 21.33% | 8.77% | 8.06% | 1.42% | 6.40% | 11.37% |
| ⑤ o3 | 10.66% | 17.46% | 9.98% | 13.15% | 17.01% | 4.31% | 1.36% | 10.43% | 15.65% |
| ⑤ o4-mini | 18.03% | 17.62% | 7.99% | 16.60% | 16.19% | 3.89% | 1.43% | 9.02% | 9.22% |
| ⑤ GPT-4o | 5.41% | 21.65% | 12.12% | 14.50% | 11.26% | 7.58% | 2.60% | 10.82% | 13.85% |
| ⑤ GPT-4.1 | 11.97% | 21.58% | 8.55% | 18.80% | 9.83% | 6.41% | 1.71% | 9.62% | 11.54% |
| ⑤ GPT-5 | 15.93% | 13.05% | 6.91% | 31.67% | 10.17% | 3.65% | 1.15% | 10.94% | 6.53% |
| ∅ Grok-4 | 3.72% | 24.07% | 17.03% | 17.81% | 10.18% | 5.68% | 2.94% | 6.46% | 12.13% |
| ᴍ Mistral-Large | 8.93% | 24.08% | 15.34% | 8.16% | 14.76% | 5.05% | 1.94% | 8.16% | 13.40% |
| ◑ Doubao-1.5 | 15.16% | 31.47% | 21.32% | 2.29% | 11.30% | 2.72% | 1.72% | 8.44% | 5.58% |
| ◑ Doubao-1.5-Thinking | 17.27% | 27.64% | 11.45% | 5.82% | 13.09% | 4.36% | 1.82% | 11.45% | 7.09% |
| ◑ Doubao-1.6 | 1.87% | 25.37% | 17.54% | 22.39% | 10.07% | 5.22% | 1.49% | 5.78% | 10.26% |
| ◑ Doubao-1.6-Thinking | 3.87% | 30.97% | 21.77% | 16.77% | 8.87% | 4.68% | 0.97% | 6.13% | 5.97% |
| *Open-Source General Models* | | | | | | | | | |
| ∞ Llama-3.3-70B-Instruct | 62.27% | 4.32% | 5.11% | 19.66% | 5.91% | 1.14% | 1.14% | 0.11% | 0.34% |
| ∞ Llama-3.3-8B-Instruct | 77.99% | 0.00% | 0.00% | 21.66% | 0.00% | 0.36% | 0.00% | 0.00% | 0.00% |
| ∞ Llama-3.3-3B-Instruct | 78.30% | 0.00% | 0.00% | 21.70% | 0.00% | 0.00% | 0.00% | 0.00% | 0.00% |
| ∞ Llama-3.3-1B-Instruct | 80.14% | 0.00% | 0.00% | 19.86% | 0.00% | 0.00% | 0.00% | 0.00% | 0.00% |
| ⩗ Qwen2.5-72B-Instruct | 12.42% | 21.59% | 8.96% | 15.07% | 12.02% | 8.76% | 2.24% | 7.54% | 11.20% |
| ⩗ Qwen2.5-32B-Instruct | 12.89% | 17.88% | 9.15% | 16.84% | 13.72% | 7.90% | 2.08% | 7.90% | 11.64% |
| ⩗ Qwen2.5-14B-Instruct | 17.42% | 15.91% | 8.14% | 19.89% | 12.50% | 8.71% | 2.08% | 6.25% | 9.09% |
| ⩗ Qwen2.5-7B-Instruct | 11.01% | 21.49% | 12.26% | 12.61% | 20.96% | 5.51% | 1.42% | 6.75% | 7.82% |
| ⩗ Qwen2.5-3B-Instruct | 14.92% | 20.66% | 10.98% | 13.28% | 18.36% | 6.23% | 0.98% | 5.25% | 9.18% |
| ⩗ Qwen2.5-1.5B-Instruct | 34.10% | 13.11% | 4.10% | 15.57% | 16.07% | 3.44% | 0.98% | 4.59% | 7.05% |
| ⩗ Qwen2.5-0.5B-Instruct | 27.92% | 11.82% | 6.55% | 20.23% | 14.96% | 2.99% | 1.00% | 6.84% | 5.70% |
| ⩗ Qwen3-32B | 10.72% | 19.59% | 10.31% | 20.21% | 16.91% | 5.57% | 1.24% | 6.80% | 8.66% |
| ⩗ Qwen3-30B-A3B | 16.77% | 14.95% | 5.25% | 30.10% | 14.34% | 4.65% | 1.41% | 4.65% | 7.88% |
| ⩗ Qwen3-14B | 12.55% | 18.72% | 5.53% | 18.51% | 14.26% | 8.09% | 1.70% | 11.70% | 8.94% |
| ⩗ Qwen3-8B | 8.29% | 27.31% | 9.60% | 6.59% | 19.40% | 7.53% | 1.69% | 9.98% | 9.42% |
| ⩗ Qwen3-4B | 15.71% | 15.13% | 6.51% | 14.37% | 18.01% | 6.90% | 1.34% | 11.69% | 10.34% |
| ⩗ Qwen3-1.7B | 14.49% | 21.12% | 8.03% | 9.42% | 24.08% | 5.24% | 1.05% | 6.63% | 9.95% |
| ⩗ Qwen3-0.6B | 14.05% | 25.80% | 8.40% | 6.56% | 29.47% | 3.82% | 1.07% | 3.97% | 6.72% |
| ⩗ Qwen3-30B-A3B-Thinking | 18.45% | 12.90% | 4.96% | 29.37% | 14.48% | 5.16% | 1.79% | 5.16% | 7.74% |
| ⩗ Qwen3-14B-Thinking | 15.03% | 17.12% | 5.64% | 19.83% | 13.99% | 5.85% | 1.88% | 10.23% | 10.23% |
| ⩗ Qwen3-8B-Thinking | 11.32% | 19.44% | 6.84% | 14.10% | 17.52% | 6.84% | 2.14% | 10.47% | 11.32% |
| ⩗ Qwen3-4B-Thinking | 12.71% | 20.55% | 6.14% | 14.41% | 17.37% | 5.72% | 3.18% | 8.69% | 11.23% |
| ⩗ Qwen3-1.7B-Thinking | 15.31% | 20.66% | 7.20% | 12.73% | 16.05% | 7.01% | 2.03% | 7.20% | 11.44% |
| ⩗ Qwen3-0.6B-Thinking | 18.26% | 19.45% | 5.63% | 15.70% | 23.04% | 4.78% | 1.02% | 4.61% | 7.51% |
| ✷ GLM-4.5 | 10.89% | 19.33% | 10.67% | 18.89% | 15.33% | 6.00% | 1.11% | 4.89% | 12.89% |
| Ⲕ Kimi-K2 | 21.31% | 13.50% | 7.17% | 16.24% | 11.60% | 6.54% | 2.53% | 6.96% | 14.14% |
| ⚭ DeepSeek-R1 | 13.54% | 14.41% | 11.14% | 20.96% | 11.79% | 6.33% | 1.31% | 8.73% | 11.79% |
| ⚭ DeepSeek-V3 | 10.52% | 21.65% | 10.93% | 15.88% | 16.49% | 5.15% | 1.44% | 7.63% | 10.31% |
| ⚭ DeepSeek-V3.1 | 20.00% | 11.76% | 9.41% | 25.29% | 13.92% | 3.73% | 0.98% | 5.69% | 9.22% |
| *Open-Source Specialized Models* | | | | | | | | | |
| ⬗ xLAM-2-70B | 6.48% | 30.67% | 17.14% | 4.38% | 16.19% | 5.71% | 0.95% | 5.90% | 12.57% |
| ⬗ xLAM-2-32B | 8.98% | 21.76% | 15.37% | 8.38% | 18.56% | 4.19% | 1.60% | 5.39% | 15.77% |
| ⬗ xLAM-2-8B | 5.60% | 27.33% | 19.86% | 3.90% | 19.52% | 5.09% | 2.21% | 4.92% | 11.21% |
| ⬗ xLAM-2-3B | 6.84% | 28.30% | 17.11% | 5.13% | 21.00% | 4.98% | 1.56% | 4.82% | 9.64% |
| ⬗ xLAM-2-1B | 9.40% | 24.61% | 13.95% | 5.96% | 22.57% | 4.86% | 1.88% | 3.61% | 10.66% |
| ▦ ToolACE2-8B | 10.11% | 28.60% | 8.60% | 6.67% | 18.28% | 6.02% | 2.80% | 4.09% | 14.84% |
| ◉ Watt-8B | 5.97% | 30.97% | 10.45% | 7.09% | 23.13% | 4.29% | 1.49% | 6.16% | 10.45% |
| ✖ Hammer2.1-7B | 38.24% | 15.81% | 2.39% | 12.68% | 15.26% | 1.84% | 0.55% | 4.41% | 8.82% |
| ✖ Hammer2.1-3B | 36.13% | 19.18% | 3.65% | 19.81% | 8.40% | 3.01% | 0.95% | 2.06% | 6.66% |
| ✖ Hammer2.1-1.5B | 60.47% | 3.29% | 1.32% | 32.15% | 1.32% | 0.26% | 0.00% | 0.26% | 0.92% |
| ✖ Hammer2.1-0.5B | 30.80% | 17.69% | 4.46% | 22.15% | 13.11% | 2.75% | 0.13% | 6.82% | 1.97% |

---

**Algorithm 1** Enumeration of All Serial and Parallel Execution Paths

---

**Require:** Directed acyclic graph $G = (V, E)$; annotated length $L$
**Ensure:** All paths $\mathcal{P}$, divided into optimal and suboptimal sets
 1: Compute indegree$[v]$ for all $v \in V$
 2: Initialize visited$[v] \leftarrow$ false for all $v \in V$
 3: CurrentPath $\leftarrow \emptyset$, $\mathcal{P} \leftarrow \emptyset$
 4: **function** ZEROINDEGREE(indegree, visited)
 5:     **return** $\{v \in V \mid$ indegree$[v] = 0 \wedge \neg$visited$[v]\}$
 6: **end function**
 7: **function** COMBINATIONS($Z$)
 8:     **return** all non-empty subsets of $Z$
 9: **end function**
10: **procedure** DFS(indegree, visited, CurrentPath)
11:     $Z \leftarrow$ ZEROINDEGREE(indegree, visited)
12:     **if** $Z = \emptyset$ **then**
13:         **if** $|$CurrentPath$| = |V| \vee |$CurrentPath$| = L$ **then**
14:             Add copy of CurrentPath to $\mathcal{P}$
15:         **end if**
16:         **return**
17:     **end if**
18:     **for all** $C \in$ COMBINATIONS($Z$) **do**
19:         Backup indegree, visited, CurrentPath
20:         Append $C$ to CurrentPath
21:         Mark all $v \in C$ as visited
22:         **for all** edges $(v, u)$ with $v \in C$ **do**
23:             indegree$[u] \leftarrow$ indegree$[u] - 1$
24:         **end for**
25:         DFS(indegree, visited, CurrentPath)
26:         Restore backup
27:     **end for**
28: **end procedure**
29: DFS(indegree, visited, CurrentPath)
30: $L^* \leftarrow \min\{|p| : p \in \mathcal{P}\}$
31: $\mathcal{P}_{opt} \leftarrow \{p \in \mathcal{P} : |p| = L^*\}$
32: $\mathcal{P}_{sub} \leftarrow \mathcal{P} \setminus \mathcal{P}_{opt}$
33: **return** $\mathcal{P}, \mathcal{P}_{opt}, \mathcal{P}_{sub}$

---

# F  PROMPTS

## F.1  PROMPT FOR SINGLE-TOOL CALLS SEED TASK GENERATION

We show the role prompt of the single-tool calls task generation in Figure 9.

## F.2  PROMPT FOR SEQUENTIAL MULTI-TOOL CALLS SEED TASK GENERATION

We show the role prompt of sequential multi-tool calls task generation in Figure 10.

## F.3  PROMPT FOR PARALLEL MULTI-TOOL CALLS SEED TASK GENERATION

We show the role prompt of parallel multi-tool calls task generation in Figure 11.

## F.4  PROMPT FOR MIXED MULTI-TOOL CALLS SEED TASK GENERATION

We show the role prompt of mixed multi-tool calls task generation in Figure 12.

### F.5 PROMPT FOR CLARIFY SEED TASK GENERATION

We show the role prompt of the clarify task generation in Figure 13.

### F.6 PROMPT FOR CHAT SEED TASK GENERATION

We show the role prompt of chat task generation in Figure 14.

### F.7 PROMPT FOR CONTEXT SEED TASK GENERATION

We show the role prompt of context task generation in Figure 15 and Figure 16.

## G ERROR CASES

Figure 17 presents several typical error examples discussed in the main text.

---

**Single-Tool Calls task Generation Prompt.**

Please act as a user interacting with a super intelligent agent.

This super intelligent agent has access to a range of external tools and can use these tools to solve the tasks you propose.

Next, please propose 5 tasks that you need the super intelligent agent to solve based on the [Requirements].

All 5 tasks must require the use of $\{\{\{tool\}\}\}$ from the [Tool List] to be completed, and each task should only require a single call to $\{\{\{tool\}\}\}$.

The tasks should be specific and diverse.

Finally, please output the final result according to the [Format] without generating any extra text.

The required parameters for tool $\{\{\{tool\}\}\}$ are: $\{\{\{tool\_required\}\}\}$, and the optional parameters are: $\{\{\{tool\_no\_required\}\}\}$.

[Requirements]="""
1. The description of the user's task must include information on all the required parameters needed to call $\{\{\{tool\}\}\}$. For other optional parameters, please add them as you see fit, using natural language.
2. The user's tasks should use different types of sentence structures: imperative, declarative, interrogative, etc.
3. The user's tasks should include different tones: colloquial, formal, polite, direct, etc.
4. Ensure that the length of the user's tasks varies, gradually increasing from short to long.
5. Ensure that the user's tasks involve different themes/instances, different scenarios, and different roles.
6. Extract common entities that appear in all descriptions from the [Tool List] and ensure that these entities appear in the user's tasks.
7. Do not explicitly specify the tool $\{\{\{tool\}\}\}$ in the user's tasks.
"""

[Tool List]="""
$\{\{\{tool\}\}\}$
"""

[Format]="""
{
    "task 1": "xxx",
    "task 2": "xxx",
    "task 3": "xxx",
    "task 4": "xxx",
    "task 5": "xxx",
}
"""

Figure 9: Single-Tool Calls task Generation Prompt.

---

**Sequential Multi-Tool Calls task Generation Prompt.**

Please act as a user interacting with a super intelligent agent.

This super intelligent agent has access to a range of external tools and can use these tools to solve the tasks you propose.

Next, based on the [Requirements], please propose 5 tasks that you need the super intelligent agent to solve.

These 5 tasks must require the combined use of tools from the [Tool List] (including: {{{all_tool_name}}}) to be completed.

The tasks should be specific, diverse, and require the sequential invocation of multiple tools to solve.

Finally, please output the final result according to the [Format] without generating any extra text.

{{{all_tool_required_info}}}

[Requirements]="""
1. The description of the user's task must include all the required parameters needed to invoke the tools, while other optional parameters can be added as you see fit, using natural language.
2. The user's tasks should use different types of sentence structures: imperative, declarative, interrogative, etc.
3. The user's tasks should include different tones: colloquial, formal, polite, direct, etc.
4. Ensure that the length of the user's tasks varies, from short to long, gradually increasing in length.
5. Ensure that the user's tasks involve different themes/instances, different scenarios, and different roles.
6. Based on the descriptions of all tools in the [Tool List], extract the common entities that appear in all descriptions and ensure that these entities appear in the user's tasks.
7. There must be dependencies between the multiple tools invoked, meaning that tool A must be called and completed before tool B can be run, i.e., tool B must be invoked after tool A.
8. The difficulty of the tasks is divided into easy, medium, and hard levels. Easy represents simple, medium represents moderate, and hard represents difficult. Ensure that the 5 tasks you generate are all of medium difficulty or above.
9. Do not explicitly specify the names of the tools to be used in the user's tasks.
"""

[Tool List]="""
{{{tools}}}
"""

[Format]="""
{
    "task 1": "xxx",
    "task 2": "xxx",
    "task 3": "xxx",
    "task 4": "xxx",
    "task 5": "xxx",
}
"""

Figure 10: Sequential Multi-Tool Calls task Generation Prompt.

---

**Parallel Multi-Tool Calls task Generation Prompt.**

Please act as a user interacting with a super intelligent agent.

This super intelligent agent has access to a range of external tools and can use these tools to solve the tasks you propose.

Next, based on the [Requirements], please propose 5 tasks that you need the super intelligent agent to solve.

These 5 tasks must require the combined use of tools from the [Tool List] (including: {{{all_tool_name}}}) to be completed.

The tasks need to be specific, diverse, and require parallel invocation of multiple tools to solve.

Finally, please output the final result according to the [Format] without generating any extra text.

{{{all_tool_required_info}}}

[Requirements]="""
1. The description of the user's task must include all the required parameters needed to invoke the tools, while other optional parameters can be added as you see fit, using natural language.
2. The user's tasks should use different types of sentence structures: imperative, declarative, interrogative, etc.
3. The user's tasks should include different tones: colloquial, formal, polite, direct, etc.
4. Ensure that the length of the user's tasks varies, from short to long, gradually increasing in length.
5. Ensure that the user's tasks involve different themes/instances, different scenarios, and different roles.
6. Based on the descriptions of all tools in the [Tool List], extract the common entities that appear in all descriptions and ensure that these entities appear in the user's tasks.
7. There must be no dependency between the multiple tools invoked. A dependency between invocations means that tool B can only be run after tool A is completed. No dependency means that tool A and tool B can be invoked in parallel.
8. The difficulty of the tasks is divided into easy, medium, and hard levels. Easy represents simple, medium represents moderate, and hard represents difficult. Ensure that the 5 tasks you generate are all of medium difficulty or above.
9. Do not explicitly specify the names of the tools to be used in the user's tasks.
"""

[Tool List]="""
{{{tools}}}
"""

[Format]="""
{
    "task 1": "xxx",
    "task 2": "xxx",
    "task 3": "xxx",
    "task 4": "xxx",
    "task 5": "xxx",
}
"""

Figure 11: Parallel Multi-Tool Calls task Generation Prompt.

Mixed Multi-Tool Calls task Generation Prompt.

Please act as a user interacting with a super intelligent agent.

This super intelligent agent has access to a range of external tools and can use these tools to solve the tasks you propose.

Next, based on the [Requirements], please propose 5 tasks that you need the super intelligent agent to solve.

These 5 tasks must require the combined use of tools from the [Tool List] (including: {{{all_tool_name}}}) to be completed.

The tasks should be specific, diverse, and require both serial and parallel invocation of multiple tools to solve.

Finally, please output the final result according to the [Format] without generating any extra text.

{{{all_tool_required_info}}}

[Requirements]="""
1. The description of the user's task must include all the required parameters needed to invoke the tools, while other optional parameters can be added as you see fit, using natural language.
2. The user's tasks should use different types of sentence structures: imperative, declarative, interrogative, etc.
3. The user's tasks should include different tones: colloquial, formal, polite, direct, etc.
4. Ensure that the length of the user's tasks varies, from short to long, gradually increasing in length.
5. Ensure that the user's tasks involve different themes/instances, different scenarios, and different roles.
6. Based on the descriptions of all tools in the [Tool List], extract the common entities that appear in all descriptions and ensure that these entities appear in the user's tasks.
7. There should be dependencies between some of the tools invoked, while others should not have dependencies. A dependency between invocations means that tool B can only be run after tool A is completed. No dependency means that tool A and tool B can be invoked in parallel.
8. The difficulty of the tasks is divided into easy, medium, and hard levels. Easy represents simple, medium represents moderate, and hard represents difficult. Ensure that the 5 tasks you generate are all of medium difficulty or above.
9. Do not explicitly specify the names of the tools to be used in the user's tasks.
"""

[Tool List]="""
{{{tools}}}
"""

[Format]="""
{
    "task 1": "xxx",
    "task 2": "xxx",
    "task 3": "xxx",
    "task 4": "xxx",
    "task 5": "xxx",
}
"""

Figure 12: Mixed Multi-Tool Calls task Generation Prompt.

---

**Clarify task Generation Prompt.**

Please act as a user interacting with a super intelligent agent.

This super intelligent agent has access to a range of external tools and can use these tools to solve the tasks you propose.

Next, based on the [Requirements], please propose 5 tasks that you need the super intelligent agent to solve.

These 5 tasks must require the combined use of tools from the [Tool List] (including: {{{all_tool_name}}}) to be completed.

All 5 tasks must require the use of {{{tool}}} from the [Tool List] to be completed, but will leave the super intelligent agent unclear on how to fill in some of the required parameters of {{{tool}}}, and should be diverse.

Finally, please output the final result according to the [Format] without generating any extra text.

The required parameters for tool {{{tool}}} are: {{{tool_required}}}, and the optional parameters are: {{{tool_no_required}}}

[Requirements]="""
1. The description of the user's task must lack all the necessary information for calling {{{tool}}}, leaving only the optional parameter information, which you can add as you see fit, using natural language descriptions. Note that tool parameters allow for some parameter inference, meaning that if the tool parameters can be inferred from the user's task description, it does not count as lacking necessary information. Lacking means that even through inference, the parameter values cannot be obtained.
2. The user's tasks need to use different types of sentence structures: imperative sentences, declarative sentences, interrogative sentences, etc.
3. The user's tasks should include different tones: colloquial, formal, polite, direct, etc.
4. Ensure that the length of the user's tasks varies, from short to long, gradually increasing in length.
5. Ensure that the user's tasks involve different themes/instances, different scenarios, and different roles.
6. Based on the descriptions of all tools in the [Tool List], extract the common entities that appear in all descriptions and ensure that these entities appear in the user's tasks.
7. Task difficulty is divided into easy, medium, and hard levels. Easy represents simple, medium represents moderate, and hard represents difficult. More difficult tasks require more steps to execute. Ensure that the 3 tasks you generate are all of medium difficulty or above.
8. Do not explicitly specify the tool {{{tool}}} in the user's tasks.
"""

[Tool List]="""
{{{tools}}}
"""

[Format]="""
{
     "task 1": "xxx",
     "task 2": "xxx",
     "task 3": "xxx",
     "task 4": "xxx",
     "task 5": "xxx",
}
"""

Figure 13: Clarify task Generation Prompt.

---

**Chat task Generation Prompt.**

Please act as a user interacting with a super intelligent agent.

This super intelligent agent has access to a range of external tools and can use these tools to solve the tasks you propose.

Next, based on the [Requirements], propose 5 casual conversation tasks that you need the super-intelligent agent to solve.

These 5 casual conversation tasks should not use any tools from the [Tool List], but should have some thematic relevance.

Finally, please output the final result according to the [Format] without generating any extra text.

The required parameters for tool {{{tool}}} are: {{{tool_required}}}, and the optional parameters are: {{{tool_no_required}}}

[Requirements]="""
1. The user task is a casual conversation task, which must be unrelated to the functions of the [Tool List], but should have some thematic relevance.
2. User tasks need to use different types of sentence structures: imperative, declarative, interrogative, etc.
3. User tasks should include different tones: colloquial, formal, polite, direct, etc.
4. Ensure that the lengths of the user tasks are different, ranging from short to long, with gradually increasing length.
5. Ensure that the user tasks involve different themes/examples, different scenarios, and different role identities.
"""

[Tool List]="""
{{{tools}}}
"""

[Format]="""
{
    "task 1": "xxx",
    "task 2": "xxx",
    "task 3": "xxx",
    "task 4": "xxx",
    "task 5": "xxx",
}
"""

Figure 14: Chat task Generation Prompt.

Context task Generation Prompt, Part 1.

Please act as a user interacting with a super intelligent agent.

This super intelligent agent has a Planner, an Agent assistant, and a range of external tools that can be used to solve the tasks you propose, as detailed in the [Tool List].

Based on the information in [Historical Conversations], you have already proposed your task, and the super intelligent agent has solved it for you.

Therefore, next, please continue to propose new tasks based on the reply from the Agent assistant in the last round of [Historical Conversations], referring to the [Turn Type Information] and [Example], and the new tasks you propose must require the use of {{{tool_number}}} tool from the [Tool List] to solve.

Finally, output according to the [Format].

{{{all_tool_required_info}}}

[Tool List]="""
{{{tools}}}
"""

[Turn Type Information]="""
{{{turn_type_info}}}
"""

Figure 15: Context task Generation Prompt, Part 1.

---

**Context task Generation Prompt, Part 2.**

When actually generating tasks, one of the following types will be substituted into the prompt placeholder {{{turn_type_info}}}.
1. Partial Information: The new task generated needs to omit some content from previous conversations, without having to state the full semantics. The omitted content can be any sentence component, including: subject, attribute, attribute value, modifier, etc.
2. Coreferential Reference: The new task generated requires reference to some content from previous conversations, which can be: 1) Ordinal reference, such as: the second point, the last point, etc. 2) Pronominal reference, such as: he, this sentence, which one, etc. 3) Vague reference, such as: xxx this model, etc.
3. Long-Range dependency: The new task generated needs to use content from previous conversations (excluding the last round), for example, something I mentioned in the first round, something I mentioned before.
"""

[Example]="""
[Historical Conversations]=***
{{{history}}}
***
[Output]=***
{{{continue_task}}}
***
"""

[Historical Conversations]="""
{{{history}}}
"""

[Format]="""
{
    "task 1": "xxx",
    "task 2": "xxx",
    "task 3": "xxx",
    "task 4": "xxx",
    "task 5": "xxx",
}
"""

Figure 16: Context task Generation Prompt, Part 2.

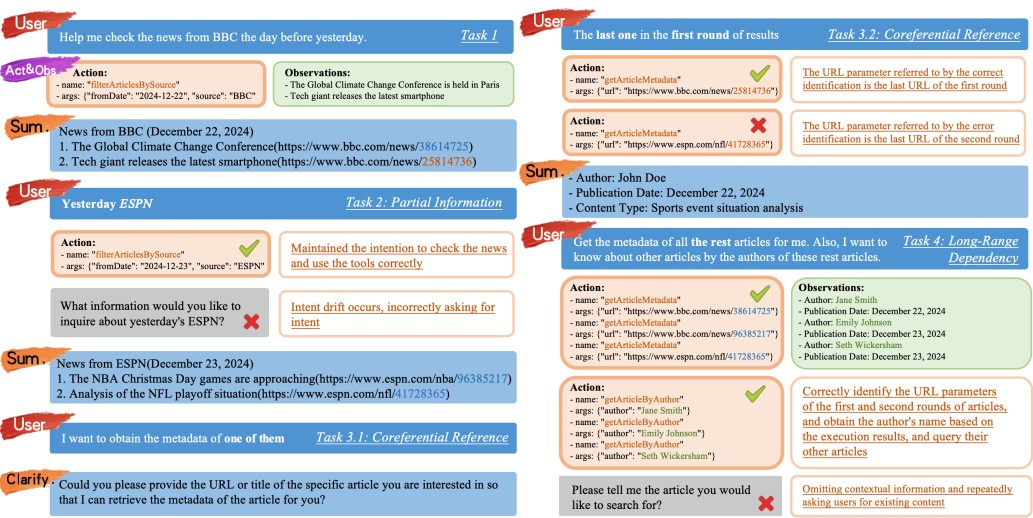

Figure 17: Typical error examples discussed in the main text

