# OpenReview forum: "Benchmarking LLM Tool-Use in the Wild"
_ICLR.cc/2026/Conference — ICLR 2026 Poster_

### Official Review · Reviewer_E7mX · 2025-10-28

**Soundness:** 3
**Presentation:** 3
**Contribution:** 3
**Rating:** 6
**Confidence:** 4

**Summary:**

This work proposes WildToolBench -- a tool-calling evaluation dataset that simulates real-user behavior when interacting with tool-calling models. This work identifies three user characteristics that LLMs find challenging, namely single tool-use, inferring information and intent from multi-turn dialog, and switching intents on the fly. This work creates 256 scenarios, with each scenario containing 4 tasks for a total of 1024 tasks. Evaluation of 57 models finds that the best performance on a session is 15%, indicating that the benchmark is hard to solve for current models.

**Strengths:**

- The underlying motivation to build a dataset mimicking realistic user conversations is a meaningful contribution. Results showing that models struggle greatly to fulfill user requirements in a session demonstrate the utility of this dataset as a good benchmark.
- Extensive experimentation covering 57 models, ranging from proprietary models to open-source general and function-calling specific models.

**Weaknesses:**

- I found the process of data curation hard to understand from the main text. The paper would benefit from reordering of the content by moving sections describing data curation from the Appendix into the main text
- Similarly, adding a section comparing the WildToolBench and other function-calling benchmarks to show how the other benchmarks fail to capture the cases that WildToolBench does will help set the motivation.
- Section 3.2 mentions that the first step to data curation is analyzing large-scale real user logs, but no information is provided about the source or the characteristics of these user logs. Similarly, the 1600 APIs collected are also not described.
- Section 4.2 title should most likely read "LLMs Perform Poorly on Tool Orchestration"?

**Questions:**

See above

---

> ### Author Response · Authors · 2025-11-20
> **Response to Reviewer E7mX**
>
> Thank you very much for your comments and for your positive review of our work. Below we respond to your concerns point by point, and please feel free to reach out if anything remains unclear.
>
> > Q1.	On formatting and typos
>
> - We are very grateful for your careful reading and helpful suggestions. We will correct the title of Section 4.2. In addition, due to space constraints, we moved the details of the data construction process to Appendix C; we will bring these details back into the main text, including Figure 8.
>
> - Due to the page limitations, we deliberately allocated more space to the construction of the three challenges and the corresponding experimental analysis, because we hope WildToolBench is not just a benchmark for leaderboard chasing, but can also serve as a “scalpel” to dissect LLM tool use in user-centric scenarios from different perspectives. As you rightly pointed out, however, this choice is not very reader-friendly. We will follow your suggestion and adjust the layout accordingly.
>
> ---
>
> > Q2. On comparing WildToolBench with other function calling benchmarks
>
> - We do compare WildToolBench with other function calling benchmarks in Table 5 in the appendix. For the same reason of space, this comparison was moved out of the main text.
>
> - We also highlight the most direct differences on the very first page of the paper, in Figure 1, where we show that WildToolBench is far from saturated and substantially more challenging than existing benchmarks, even though its task form is quite simple.
>
> - As you noted, the current main text only includes case studies that illustrate the characteristics of WildToolBench itself (Figures 3 and 4). We will add a case study in the main text showing why other benchmarks fail to capture such challenges, for example, in the $\tau$-bench, a retail example is:
>
> 	-	system: # Retail Agent Policy ... You should only make one tool call at a time ... **(The system prompt in $\tau$-bench indicates sequential tool call only)**
>
> 	-	...(Omit the previous conversation.)
>
> 	-	assistant:  I found your recent orders. Here are the details: 1. Order ID: #W9583042; 2. Order ID: #W4082615. Please let me know which order you would like me to check the address for.
>
> 	-	user: I can't remember exactly which one, but it's the jigsaw that I ordered recently. Could we look into that one, please?
>
> 	-	assistant: Function(arguments='{"order_id": "#W9583042"}', name='get_order_details')
>
> 	-	assistant: Function(arguments='{"order_id":"#W4082615"}', name='get_order_details') **(Due to constraints in the system prompt, the assistant can’t call tools in parallel and has to make two sequential tool calls, which prevents testing the model’s tool orchestration efficiency.)**
>
> 	-	...(Omit part of the conversation.)
>
> 	-	user: Yes, I'm considering changing the jigsaw I ordered. Could you tell me what is the easiest jigsaw available, like one with the least pieces fit for a young kid?
>
> 	-	assistant: The easiest jigsaw puzzle available, suitable for a young kid, is the one with the least pieces. Here are the options with 500 pieces: 1.**Item ID: 1096508426** - **Theme: Art** ... 2.Item ID: 9237024510 - Theme: Animals ...... Would you like to exchange your current jigsaw puzzle for one of these? If so, please confirm which one and provide a payment method to handle any price difference.
>
> 	-	user: Could we go with the **Art** themed one, **Item ID: 1096508426**? I generally use PayPal for my purchases. **(User’s instruction explicitly includes the ID, which is unrealistically precise and does not match how users typically phrase requests in real-world scenarios.)**
>
> ---
>
> > Q3. On the source of logs and APIs
>
> - Our user logs come from internal enterprise data, so we are unable to release the raw logs directly. This is exactly why we need a multi-agent pipeline to synthesise data on top of the raw logs for deidentification. However, our prompts are built by uniformly sampling from the user logs for few shot examples, so the generated data follow the same distribution as the original logs at the statistical level.
>
> - Figure 5 in the main paper already presents a range of statistics. It would be very helpful if you could let us know which particular characteristics of the user logs you are most interested in; we will provide as much as we can within the constraints of our privacy policy.
>
> - The source of the APIs is described in Section 3.2 and Appendix C. Our API collection follows ToolAlpaca[1], which is why we did not repeat the details at length. Concretely, the APIs come from the public and fully transparent list at https://github.com/public-apis/public-apis, which anyone can access for easy reproducibility. This GitHub repository is continuously updated and now contains more than 1400 tool lists, but to stay consistent with ToolAlpaca we use 400 of these tool lists, covering around 1600 APIs in total.
>
> reference
>
> 1. ToolAlpaca: Generalized Tool Learning for Language Models with 3000 Simulated Cases

---

> ### Author Response · Authors · 2025-11-26
>
> Dear reviewer E7mX, we sincerely thank you for your valuable comments and suggestions.
>
> Please let us know whether your concerns have been resolved; if there are any remaining issues, please do not hesitate to reply and we will provide a thorough response as soon as possible.
>
> We look forward to your reply~

---

### Official Review · Reviewer_6GJ2 · 2025-10-29

**Soundness:** 3
**Presentation:** 2
**Contribution:** 2
**Rating:** 2
**Confidence:** 4

**Summary:**

This paper introduces WildToolBench, a benchmark designed to evaluate Large Language Models' (LLMs) ability to effectively use tools in real-world, multi-turn, multi-step dialogue settings. Unlike existing benchmarks that focus on artificially complex tasks, WildToolBench addresses the complexity of user behavior in real-world interactions. The authors identify three key challenges that current LLM tool-use capabilities fail to handle effectively: compositional tasks that demand complex tool orchestration, implicit intention spread across dialogue turns, and frequent transitions between task queries, clarifications, and casual conversation. The paper benchmarks 57 LLMs, revealing that none of the models achieve higher than 15% accuracy in session completions, underscoring a significant gap in LLMs' agentic capabilities. The experiments and analysis demonstrate that LLMs struggle not with artificially difficult tasks, but with the "wild" nature of real user interactions.

**Strengths:**

By focusing on multi-turn, multi-step tool-use that mimics actual user interactions, the benchmark is highly relevant to practical applications of LLMs in domains requiring tool integration. The insights provided from this evaluation have the potential to inform the development of more robust, user-aware AI systems. Meanwhile, this paper effectively highlights the specific areas where LLMs currently struggle, such as handling compositional tasks, inferring implicit user intentions, and adapting to frequent task transitions. These findings are essential for guiding future model development and addressing fundamental weaknesses in tool-use.

**Weaknesses:**

Although the paper introduces a useful benchmark, there is a lack of transparency regarding the usage api/document. The author didn't provide any scratch version of the benchmark, which limits the reproducibility and potential impact of the paper on the community.
Additionally, a significant portion of the benchmark creation was done manually, which limits its scalability and increases costs. The current manual curation process is resource-intensive, making it difficult to expand the dataset further without significant additional effort.

**Questions:**

referred to weakness paragraph.

---

> ### Author Response · Authors · 2025-11-20
> **Response to Reviewer 6GJ2**
>
> Thank you very much for your positive assessment of our work. However, we find the weaknesses you raised a bit confusing. Below we provide a more detailed explanation and we would greatly appreciate it if you could let us know whether this addresses your concern.
>
> ---
>
> > Q1.	lack of transparency regarding the usage api/document. Provide no scratch version of the benchmark. Limited reproducibility.
>
> -  **In both the main text and the appendix, we repeatedly emphasise the source, transparency, and reproducibility of the entire data construction pipeline**, including:
> 	- Section 3.2 of the main paper explains that the APIs follow ToolAlpaca [1] and are sourced from https://github.com/public-apis/public-apis, which contains 1600 publicly available APIs. The source is fully open and completely transparent.
> 	- Section 6 of the main paper (**with the section name Reproducibility Statement**) states that all data are provided in the submission supplementary materials, which also include all API documents and descriptions.
> 	- Figure 5 in the main paper presents detailed statistics and visualisations of the APIs.
> 	- Due to space limits in the main paper, we further elaborate on the data construction pipeline in Appendix C, where we again clarify that the APIs follow ToolAlpaca and are sourced from https://github.com/public-apis/public-apis. This repository is continuously updated and now contains more than 1400 tool lists, but in order to stay consistent with ToolAlpaca[1] we use a subset of 400 tool lists, covering around 1600 APIs in total.
> 	- Our supplementary code includes detailed instructions for reproducing all results. If for any technical reason you are unable to access the ICLR submission attachment, you can refer to the anonymous repository https://anonymous.4open.science/r/WildToolBench-Anonymous-A22F/README.md, whose timestamp is " Last Update Sep 24, 2025", which shows that we have provided fully reproducible code from the time of submission.
>
> ---
>
> > Q2.	The current manual curation process is resource intensive, making it difficult to expand the dataset further without significant additional effort.
>
> - To begin with, the pipeline itself is fully automated. It is not a manual curation process. Human effort is mainly used to perform high-standard checks and fix mistakes. This is indispensable for any benchmark. We cannot rely entirely on synthetic data to construct a benchmark. **No matter how much we automate and synthesise training data, we still need a high-quality, human verified benchmark to faithfully and objectively measure model performance.**
>
> - Regarding whether this pipeline can be used to construct synthetic training data, the answer is yes, although this is beyond the scope of a benchmark paper. The pipeline can be scaled up by removing the manual checking stage, which yields an initial pool of reasonably high-quality cold start data for warm-up. This is exactly how we use it in industry. In fact, the pipeline was originally designed as a framework for synthesising training data, and we then applied strict human evaluation to select a subset as a benchmark. The specific ways this pipeline is used in training are beyond the scope of this benchmark paper, so they are not discussed in the manuscript.
>
> - It is also important to note that any strong LLM is trained using a systematic data strategy. It is impossible to rely on a single pipeline to generate the entire training corpus. Once manual checking is removed, the synthetic data will inevitably contain some errors, but it is still very suitable as an intermediate cold start stage.
>
> - Our criteria for manual checking are multi-dimensional and go far beyond verifying tool call accuracy. In fact, even without human checking, the correctness of tool calls alone is already above 85\%, which is sufficient for cold start training data. Our manual checks focus mainly on three aspects:
> 	- making the queries sound more like they are written by real users, which is the main source of edits;
> 	- enriching each scenario so that all types of task switching are fully covered;
> 	- correcting incorrect single-step tool calls, which only account for a small fraction of edits.
>
> - Consequently, most of the human edits occur in the multi-agent communication part, where we improve the naturalness of the language generated by the user-side LLM. A recent line of work has begun to study user LLMs explicitly [2,3], and we believe that user simulation with LLMs will continue to improve. Our pipeline will naturally benefit from advances in this area, further reducing the amount of human intervention required.
>
> reference
>
> 1. ToolAlpaca: Generalized Tool Learning for Language Models with 3000 Simulated Cases
>
> 2. UserRL: Training Interactive User-Centric Agent via Reinforcement Learning
>
> 3. The Era of Real-World Human Interaction: RL from User Conversations

---

> ### Author Response · Authors · 2025-11-26
>
> Dear reviewer 6GJ2, we sincerely thank you for your valuable comments and suggestions.
>
> Please let us know whether your concerns have been resolved; if there are any remaining issues, please do not hesitate to reply and we will provide a thorough response as soon as possible.
>
> We look forward to your reply~

---

### Official Review · Reviewer_Cyfy · 2025-10-31

**Soundness:** 3
**Presentation:** 4
**Contribution:** 2
**Rating:** 6
**Confidence:** 4

**Summary:**

The paper presents WildToolBench, a benchmark designed to evaluate LLMs’ real-world tool-use abilities. Built from 256 human-verified scenarios reflecting authentic user behaviors—compositional tasks, implicit intents, and instruction transitions—it exposes key weaknesses in current LLMs’ reasoning and planning. Benchmarking 57 models shows none exceed 15% session accuracy, revealing significant gaps in robustness and adaptability.

**Strengths:**

1. The paper is highly original in redefining LLM tool-use benchmarking through real-world user behavior modeling, capturing compositionality, implicit intent, and instruction transitions that prior datasets overlook.

2. The study demonstrates strong methodological rigor and clear exposition, combining large-scale human-verified data with systematic evaluation of 57 models to yield reproducible and insightful findings on LLM agent robustness.

**Weaknesses:**

1. Heavy reliance on human-verified curation (256 scenarios / 1,024 tasks) constrains scale and may bias domain coverage; moreover, this limitation implies that such a data-construction pipeline is difficult to extend for model training or optimization purposes.

2. The evaluation emphasizes function-call correctness and orchestration (OP/AP) but underplays live constraints—API latency, rate limits, outages, and cost; adding live (or high-fidelity simulated) tool execution with cost/latency/robustness metrics and failure-injection studies would better reflect real deployment trade-offs and stress agent reliability.

**Questions:**

1. In Appendix C, the paper mentions that behavior patterns were summarized by analyzing real user logs — does this imply a potential information bottleneck, where instead of fully leveraging authentic user dialogues and scenarios, the authors mainly relied on manual summarization combined with LLM synthesis?

2. The same appendix states that human experts conducted four iterative rounds to raise data quality from 62% to 100%. How exactly was this data quality measured, and what were the main issues identified during the intermediate stages?

---

> ### Author Response · Authors · 2025-11-20
> **Response to Reviewer Cyfy, Part I**
>
> Thank you very much for your comments and positive review of our work! Below we respond to your concerns point by point, and we would be very happy to discuss if anything remains unclear.
>
> ---
>
> > Q1.	Reliance on human verified curation constrains scale and may bias domain coverage
>
> A1:
>
> - For any benchmark, manual inspection is indispensable. We deliberately adopt strict standards and extensive human involvement in order to build a high-quality benchmark that faithfully measures model performance, which is precisely the role a benchmark should play.
>
> - In terms of benchmark scale, WildToolBench is not small for a benchmark, and it offers broad domain coverage. We provide a statistical comparison with commonly used benchmarks for LLM tool use [1,2,3] to illustrate this.
>
> | Benchmark    | Number of Scenarios | Number of Tasks |
> |-------------|-------------------|-----------------|
> | BFCL-V3    | 8                 | 2000+           |
> | $\tau$-Bench    | 2                 | 165             |
> | $\tau^2$-Bench   | 3                 | 278             |
> | WildToolBench | 256             | 1024            |
>
> - As shown in the statistics, WildToolBench covers a broad range of scenarios and domains, whereas previous benchmarks span only a limited number of scenarios but contain many tasks within each scenario. This lowers their difficulty: LLM developers can simply optimise for a small set of scenarios, and the large number of test cases in each scenario makes it easy to tune models specifically for those settings. For example, on the latest $\tau^2$-Bench, leading proprietary models such as Claude and GPT already achieve accuracy above 80\%, and some subtasks even reach 98\% accuracy, which suggests that the benchmark is quickly saturating.
>
> ---
>
> > Q2.	Such a data construction pipeline is difficult to extend for model training or optimisation purposes
>
> A2:
>
> -  This pipeline can be scaled up by simply removing the manual checking phase, yielding an initial pool of reasonably high-quality cold start data for warm-up. This is, in fact, how we use it in the industry. The fact is that the pipeline was originally designed as a framework for synthesising training data, and we then applied strict human evaluation to a subset of the data to carve out a benchmark. Detailing how this pipeline is used for training lies beyond the scope of a benchmark paper, which is why it is not described in the manuscript.
>
> -  It is also worth emphasising that any industry-level LLM is trained through a systematic data strategy, rather than relying on a single pipeline for the entire training corpus. Once manual checking is removed, the synthetic data will inevitably contain some errors, but it is still perfectly adequate as an intermediate cold start stage.
>
> -  The criteria we use for manual checking are multi-dimensional and go far beyond verifying tool call accuracy. In the absence of human checking, the correctness of tool calls alone is already above 85\%, which is sufficient for a cold start. Our manual checks focus mainly on three aspects:
> 	- making the queries sound more like they were written by real users, which is the main source of edits;
> 	- enriching each scenario so that all relevant types of policy switch are covered;
> 	- correcting wrong tool calls, which accounts for only a small fraction of the edits.
>
> - Consequently, most of the manual refinement happens in the multi-agent communication part, where we improve the naturalness of the language produced by the user LLM. A recent line of work [4,5] focuses on optimizing user LLMs explicitly, and we expect LLM-based user simulation to keep improving. Our pipeline will naturally benefit from these advances and will require progressively less human intervention over time.
>
> reference
>
> 1. The Berkeley Function Calling Leaderboard (BFCL): From Tool Use to Agentic Evaluation of Large Language Models.
>
> 2. $\tau$-bench: A Benchmark for Tool-Agent-user Interaction in Real-world Domains.
>
> 3. $\tau^2$-Bench: Evaluating Conversational Agents in a Dual-control Environment.
>
> 4. UserRL: Training Interactive User-Centric Agent via Reinforcement Learning
>
> 5. The Era of Real-World Human Interaction: RL from User Conversations

---

> ### Author Response · Authors · 2025-11-20
> **Response to Reviewer Cyfy, Part II**
>
> > Q3. The evaluation emphasizes function call correctness and orchestration (OP or AP) but underplays live constraints
>
> - Thank you for your suggestions, but you mentioned a variety of metrics, not only live constraints, including cost, stability, and even aspects of red teaming. These are indeed important metrics for real world deployment, and we monitor them in our production systems as well. However, we are not entirely sure how they should be viewed as indicators of LLM tool use capability. If you could further clarify your point, we would be happy to provide a more targeted response.
>
> 	- If your question concerns the real time nature of tool APIs, that is, why we do not report statistics from live tool calls, the reason is that our tool calls are served from cached results rather than from a live back end. Using cached tool calls is a very common design choice in tool benchmarks. Earlier real time designs proved hard to maintain, did not scale well, and introduced additional instability from the API providers that would confound the evaluation results[6,7].
>
> 	- If your question is why we focus on tool call correctness rather than efficiency, please note that OP, as defined in Section 3.3, is not only a correctness metric but also reflects efficiency and cost. When OP is low, the model has poor ability to orchestrate tools in parallel, which leads to lower execution efficiency and higher cost. For example, in the appendix, Table 7 shows that, o3, o4 mini, and Claude 3.7 Sonnet achieve reasonable Task Accuracy, but their OP scores are close to zero, indicating that they execute tools correctly but with very low efficiency.
>
> ---
>
> > Q4. A potential information bottleneck
>
> - Our description in the paper may have caused some confusion. We do not first summarise three main challenges from user logs and then only use these challenges as prompts to generate all data. On the contrary, WildToolBench does not suffer from an information bottleneck. Its task types are complete, and its scenario distribution matches that of real user logs:
> 	- We manually check and rewrite the data to ensure that all possible patterns under each challenge are covered, such as single turn, serial multi turn, and parallel multi turn tool calls.
> 	- We uniformly sample from real user logs and use these samples as few shot examples in the prompts, so that the generated scenarios follow the same distribution as the real logs.
>
> - The three challenges are introduced only to support controlled analyses of the experiments from different perspectives and do not constrain or distort the scenario distribution.
>
> ---
>
> > Q5.	How exactly was this data quality measured and improved from sixty two percent to one hundred percent
>
> - As discussed above, our iterative data refinement focuses on three aspects: making queries more human like, ensuring task type completeness, and correcting tool calls. Among these, query humanisation is the dominant factor. Even though we provide few shot examples, the LLM does not always follow them well, and the resulting multi-turn queries can deviate from natural human conversational patterns. That's why we need human correction. Below we provide two examples of LLM generated dialogues that were not sufficiently human like and the corresponding human rewrites:
>
> 	- Turn 1 Question: “What is the weather like in Beijing today?” → Turn 1 Answer: “Sunny, twenty five degrees” → Turn 2 Question: “What is the weather like in Shanghai today?”. After manual rewriting, this becomes: Turn 1 Question: “What is the weather like in Beijing today?” → Turn 1 Answer...... → Turn 2 Question: “How about Shanghai?”
>
> 	- Turn 1 Question: “What are the latest news stories about the NBA?” → Turn 1 Answer: “1. NewsID 10001, title Luka Dončić traded to the Lakers; 2. NewsID 10002, title Mavericks get the first overall pick” → Turn 2 Question: “What is the detailed content of the first news item with NewsID 10001?”. After manual rewriting, this becomes: Turn 1 Question: “What are the latest news stories about the NBA?” → Turn 1 Answer...... → Turn 2 Question: “What is the detailed content of the first news?"
>
> Thank you again for these helpful suggestions. We will further improve Appendix C to provide a more detailed description of the data synthesis pipeline, along with the full manual for human checking.
>
> reference
>
> 6. Toolllm: Facilitating large language models to master 16000+ real-world apis
>
> 7. Stabletoolbench: Towards stable large-scale benchmarking on tool learning of large language models

---

> ### Author Response · Authors · 2025-11-26
>
> Dear reviewer Cyfy, we sincerely thank you for your valuable comments and suggestions.
>
> Please let us know whether your concerns have been resolved; if there are any remaining issues, please do not hesitate to reply and we will provide a thorough response as soon as possible.
>
> We look forward to your reply~

---

### Meta-Review · Area_Chair_DqSY · 2026-01-06

**Summary:**

All the reviewers agreed on the high quality and the importance of the proposed benchmark. The major concerns include -

1. The benchmark creation was heavily dependent on manual inspection, which hinders scaling up (Reviewers Cyfy and 6GJ2)
2. Evaluation only focused on function call correctness, ignoring other aspects and live challenges (Reviewer Cyfy).
3. The lack of transparency of the relevant sources from which the benchmark is developed, which impacts reproducibility (Reviewers 6GJ2 and E7mX).
4. The paper is hard to understand as too many details are moved to the appendix (Reviewer E7mX).

**Reviewer Concerns:**

Most concerns have been adequately addressed. In particular -

1. The authors pointed out that manual creation is a necessary step to ensure the high quality of the benchmark. They also mentioned that the proposed pipeline can be adapted to a scalable training pipeline simply by removing the manual checking steps.
2. The authors clarified that the proposed OP does not only account for tool call correctness, but also efficiency and cost.
3. The authors pointed out multiple places in the paper that provide details of the dataset creation sources. The authors also modified the paper to move important details back into the main paper.
4. The authors modified the paper to fix the typos and move important details into the main paper.

Since significant details are provided in the appendix, I would encourage the authors to ensure that the details are adequately referenced in the main text.

**Reviewer Scores:**

Reviewer Cyfy may maintain their score to 6 because their concerns about live challenges are not fully addressed.

Reviewer 6GJ2 may increase the score. The original score is a little underrated. A fair score, after the rebuttal, should be 6.

Reviewer E7mX may maintain their score to 6, as the current version still does not contain sufficient reference to the appendix.

After the rebuttal, the paper becomes a borderline paper leaning towards acceptance. Considering the high quality and the significance of the proposed benchmark. I would recommend accept. The authors are encouraged to further improve the clarity of the paper.

---

### Decision · Program_Chairs · 2026-01-26

Accept (Poster)